# Efficient protein production inspired by how spiders make silk

Nina Kronqvist[1], Médoune Sarr[1], Anton Lindqvist[2], Kerstin Nordling[1], Martins Otikovs[3], Luca Venturi[4], Barbara Pioselli[4], Pasi Purhonen[5], Michael Landreh[6], Henrik Biverstål[1,3], Zigmantas Toleikis[3], Lisa Sjöberg[1], Carol V. Robinson[6], Nicola Pelizzi[4], Hans Jörnvall[7], Hans Hebert[5], Kristaps Jaudzems[3], Tore Curstedt[8], Anna Rising[1,9] & Jan Johansson[1,9,10]

Membrane proteins are targets of most available pharmaceuticals, but they are difficult to produce recombinantly, like many other aggregation-prone proteins. Spiders can produce silk proteins at huge concentrations by sequestering their aggregation-prone regions in micellar structures, where the very soluble N-terminal domain (NT) forms the shell. We hypothesize that fusion to NT could similarly solubilize non-spidroin proteins, and design a charge-reversed mutant (NT*) that is pH insensitive, stabilized and hypersoluble compared to wild-type NT. NT*-transmembrane protein fusions yield up to eight times more of soluble protein in *Escherichia coli* than fusions with several conventional tags. NT* enables transmembrane peptide purification to homogeneity without chromatography and manufacture of low-cost synthetic lung surfactant that works in an animal model of respiratory disease. NT* also allows efficient expression and purification of non-transmembrane proteins, which are otherwise refractory to recombinant production, and offers a new tool for reluctant proteins in general.

[1] Division for Neurogeriatrics, Department of NVS, Center for Alzheimer Research, Karolinska Institutet, 141 57 Huddinge, Sweden. [2] Spiber Technologies AB, 106 91 Stockholm, Sweden. [3] Latvian Institute of Organic Synthesis, Department of Physical Organic Chemistry, Riga 1006, Latvia. [4] Chiesi Farmaceutici, R&D Department, Largo Belloli 11/A, IT-43122 Parma, Italy. [5] Department of Biosciences and Nutrition, Karolinska Institutet, and School of Technology and Health, KTH Royal institute of Technology, 141 83 Huddinge, Sweden. [6] Department of Chemistry, Physical and Theoretical Chemistry Laboratory, University of Oxford, South Parks Road, Oxford OX1 3QZ, UK. [7] Department of Medical Biochemistry and Biophysics, Karolinska Institutet, 171 77 Stockholm, Sweden. [8] Department of Molecular Medicine and Surgery, Karolinska Institutet at Karolinska University Hospital, 171 76 Stockholm, Sweden. [9] Department of Anatomy, Physiology and Biochemistry, Swedish University of Agricultural Sciences, Box 7011, 750 07 Uppsala, Sweden. [10] School of Natural Sciences and Health, Tallinn University, 101 20 Tallinn, Estonia. Correspondence and requests for materials should be addressed to N.K. (email: nina.kronqvist@ki.se).

Membrane-associated proteins account for 20–30% of the proteome[1] and are the targets of ∼60% of currently available pharmaceutical drugs[2]. To get sequestered into the membrane, a protein needs at least one stretch of 15–20 amino acid residues that promotes membrane insertion[3]. At the same time, hydrophobicity of the amino acid side chains is an important determinant of aggregation potential[4]. Hydrophobic amino acid residues also promote β-sheet formation and are overrepresented in amyloid forming core regions of many disease associated proteins[5]. Accordingly, transmembrane (TM) proteins are prone to aggregate, which may severely impede or even prevent the production of functional recombinant proteins. To circumvent this problem, several amphiphilic membrane-mimicking or micelle-forming compounds have been developed to stabilize TM proteins in aqueous solutions, for example, small-molecule detergents, protein based nanodiscs[6], or amphiphilic polymers[7] and peptides[8–11]. An alternative is to express the desired protein or peptide in fusion with a solubility enhancing protein domain that supports correct folding and promotes solubility to its fusion partner. Solubility tags are typically removed by proteolysis but can also be maintained integrated with the protein to ensure functionality during downstream characterization. Numerous fusion partners have been reported, and although some have been more successful, they must be evaluated empirically in each case[12]. Thioredoxin (Trx), maltose-binding protein (MBP), glutathione S-transferase (GST) and ubiquitin (Ub) are among the most widely used solubility tags that accumulate to high levels in the E. coli cytoplasm and have proven to markedly increase the solubility of many heterologous proteins[13–15]. The immunoglobulin-binding domain B1 from Streptococcal protein G (PGB1) is another well investigated fusion tag that allows soluble expression of various small proteins and peptides and can remain integrated during downstream structural characterizations due to its small size[16,17]. Staphylococcal nuclease A (SN), intestinal fatty acid-binding protein (IFABP) and a 19 repeat tetrapeptide sequence (NANP)$_{19}$ are examples of less examined tags that have allowed expression of integral TM and β-amyloid peptides[18–21].

The performance of a solubility tag is dependent on a number of qualities, including expression level and the ability to mediate correct folding and solubility to the target protein. We hypothesized that an N-terminal hydrophilic domain derived from spider silk protein could be exploited for expression of recombinant proteins based on its natural function. Spider silk consists mainly of large and aggregation-prone proteins (spidroins) that are produced in abdominal glands of spiders[22]. Spidroins from ampullate glands are built up from extensive stretches of repeated alanine- and glycine-rich segments flanked by globular and hydrophilic N- and C-terminal domains. During spinning, spidroins are passaged through a narrowing duct and convert into solid fibres in a process that involves precise control of the environmental conditions[23]. Despite their aggregation-prone nature, spidroins are stored at remarkably high concentrations (30–50% w/w) in the spider silk gland[24,25]. Studies propose that the unusually high solubility is attributed to the amphiphilic nature of spider (and silkworm) silk proteins, allowing them to arrange into micellar structures with the hydrophilic terminal domains sequestering the more hydrophobic repeat regions from the aqueous surrounding, thus preventing premature β-sheet formation[26,27] (Fig. 1a). The highly conserved N-terminal domain (NT) folds into a soluble ∼130 residue 5-helix bundle with a dipolar charge distribution[28,29] and forms antiparallel dimers at a pH below 6.5 (refs 30–35), thus interconnecting spidroins in the spinning duct (Fig. 1a). Investigations on recombinant spidroins derived from the Euprosthenops australis major ampullate spidroin protein

(MaSp) 1 revealed that NT mediates solubility in its monomeric conformation at neutral and slightly basic pH[29], that is, the condition at which native spidroins are stored in the gland[23]. In addition, recombinant NT is expressed at remarkably high levels in E. coli and could be concentrated to ∼216 mg ml$^{-1}$ (ref. 36).

To investigate if fusion to NT enables heterologous production of TM peptides and aggregation-prone proteins, we sought for pharmaceutically relevant peptides and proteins that previously have been difficult to produce recombinantly due to hydrophobicity and/or propensity to aggregate or fibrillate during expression. The panel we identified includes the surfactant protein analogues SP-C33Leu[37], KL4 (ref. 38) and SP-C$_{ss}$ (ref. 39); fragment human surfactant protein D (fhSP-D)[40]; cholecystokinin-58 (CCK-58)[41]; amyloid β-peptides Aβ1-40 and Aβ1-42 (ref. 42); human antimicrobial cathelicidin LL-37 precursor protein (hCAP18)[43] and a designed β-sheet protein (β17)[44]. TM peptides are mainly represented by the surfactant protein C (SP-C) analogues, which are strictly hydrophobic and highly aggregation-prone, although the amyloid β-peptides and the LL-37 peptide from hCAP18 also contain hydrophobic segments that associate to membranes.

SP-C33Leu, KL4, fhSP-D and CCK-58 were here studied in more detail. SP-C is produced by alveolar type II cells and is a constituent of surfactant, which is necessary to prevent alveolar collapse at end expiration. Mature SP-C is a TM α-helical lipopeptide of 4.2 kDa[45,46], perhaps the most hydrophobic peptide isolated from mammals. Premature infants often suffer from respiratory distress syndrome (RDS) due to insufficient amounts of surfactant. Today, this condition is treated with surfactant preparations extracted from animal lungs, for example, Curosurf, Infasurf, Alveofact and Survanta. Treatment with exogenous surfactant is also potentially beneficial for adult patients with respiratory distress, but clinical trials have so far been disappointing[47]. Surfactant preparations based on peptides produced in a heterologous system would be a possible alternative to the natural extracts used today (and formulations containing chemically synthesized peptides) and would also allow efficient screening of structure activity relationships for new analogues. However, recombinant production of SP-C has been notoriously difficult because of its extremely hydrophobic nature[48]. Successful attempts include recombinant bacterial production of SP-C analogues with the two Cys residues exchanged for Ser (herein referred to as SP-C$_{ss}$) or Phe, and in fusion with bacterial chloramphenicol acetyl transferase (CAT)[39] or SN[21], respectively. However, CAT fusion requires refolding from inclusion bodies, resulting in less active peptide, while SN fusion gives low yields that would be inadequate for scaled-up manufacturing. SP-C33Leu is an SP-C analogue, developed after more than two decades of structure activity studies of various synthetic SP-C analogues[37,49–51]. KL4 is another surfactant protein analogue designed to imitate the properties of the lung surfactant protein B (SP-B) and consists of iterated repeats of Lys-Leu-Leu-Leu-Leu[38]. SP-C33Leu and KL4 recapitulate the function of native surfactant peptides, including transmembraneous insertion[52,53], but are less prone to aggregate[49] and are therefore feasible to produce for development of synthetic surfactant preparations. KL4 surfactant is approved by the FDA for prophylactic treatment of premature infants[54,55], and CHF5633 based on chemically synthesized SP-C33Leu is in clinical trials for neonatal RDS[56].

The hydrophilic surfactant proteins A (SP-A) and D (SP-D) belong to the collectin family of proteins and are innate immune proteins participating in the pulmonary host defence system against pathogens and allergens. SP-D is proposed to have a protective role against various lung diseases[57] and much effort has been made to produce recombinant variants of SP-D with the

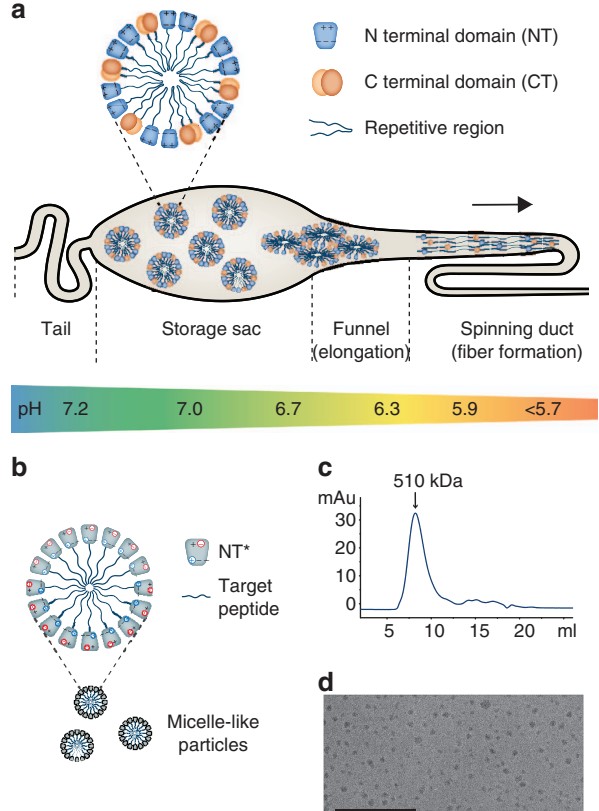

**Figure 1 | Spider silk proteins are stored as micellar structures—rationale of the current approach.** (**a**) Spidroins are synthesized in the tail region of the gland and stored at physiological pH in the sac. Premature aggregation may be prevented by formation of micelles, where the hydrophilic NT and CT domains sequester the hydrophobic and repetitive regions. The dipolar charge distribution of NT is illustrated by $+$ and $-$ signs. During passage through the spinning duct, the pH is gradually lowered, resulting in interconnection of spidroins through antiparallel dimerization of the NT domains. Assembly of spidroins gradually takes place due to changes in environmental factors and shear forces along the narrowing duct. At the exit point, the repetitive regions have arranged into strong fibres consisting of mainly β-sheets, making the spidroins inherently aggregation prone. (**b**) Recombinant production of hydrophobic and/or aggregation-prone peptides or proteins can be achieved using NT as a fusion tag that mediates solubility and shields hydrophobic/aggregation-prone regions from the aqueous surrounding within micelle-like particles. The mutant NT* is unable to dimerize at low pH due to a reduced dipolar charge distribution and is therefore able to mediate solubility in a wider pH range than NT$_{wt}$ (**c**) Size-exclusion chromatography of NT*-rSP-C33Leu shows that that the purified amphipathic fusion protein arrange into 510 kDa assemblies and (**d**) micelle-like particles around 10–15 nm in size are observed by TEM. Scale bar, 200 nm.

purpose of investigating their therapeutic potential[58,59]. Native human SP-D comprises four domains: a cysteine-linked N-terminal domain, a triple-helical collagen domain composed of Gly-Xaa-Yaa repeats, an α-helical coiled-coil domain (neck) and a globular C-terminal carbohydrate-recognition domain (CRD). Three protein chains associate through their neck domains to form active trimeric subunits that further assemble to larger multimers[60]. Heterologous expression of full-length SP-D has so far only been successful in mammalian expression systems, while truncated forms have been evaluated for expression in yeast and bacteria[59]. Recombinant fragment human (rfh)SP-D corresponds to the CRD, the neck and a

short stalk of the collagen domain[40,61]. RfhSP-D has been expressed in *E. coli* but required time-consuming and inefficient solubilization in denaturing agents and subsequent refolding[40].

Cholecystokinin (CCK) is a peptide hormone involved in the digestion process and appetite regulation and is released as a precursor peptide (pro-CCK) from cerebral neurons and endocrine I-cells in the small intestine[62]. Multiple forms exist as a result of cell-specific posttranslational processing of pro-CCK into peptide fragments ranging from 8 to 58 residues and CCK-58 has been reported as one of the main forms present in human intestine and circulation[41]. Human pro-CCK has previously been expressed at low yields in yeast[63] but recombinant production of human CCK-58 (rCCK-58), has not been reported. To obtain significant quantities of peptide for structural determination and receptor-binding studies, it would be beneficial to use bacterial expression.

In this work, we investigate the biophysical properties of a designed NT mutant and show that it enables recombinant production of a panel of hydrophobic and/or aggregation-prone peptides and proteins with higher yields compared to several commonly used solubility tags. We further show that the surfactant protein analogue SP-33Leu can be produced without the use of chromatography and that the peptide is functional in an *in vivo* model of RDS.

## Results

**Design and characterization of a charge-reversed mutant NT*.** Considering that wild-type NT (NT$_{wt}$) confers solubility primarily in its monomer conformation above pH 6.5, we designed a charge-reversed double mutant with the intention to increase the useful pH range by preventing dimerization, while still maintaining the ability to arrange into micelle-like particles in fusion with a TM peptide (Fig. 1b), in agreement with how the non-polar parts of spidroins are protected within micellar structures (Fig. 1a). The NT dimerization process is highly dependent on intermolecular electrostatic interactions between the residues Asp40 and Lys65 that play key roles in the initial association of monomers[32–35]. We designed the double mutant NT$_{D40K/K65D}$ (herein referred to as NT*) by replacing Asp40 with Lys and Lys65 with Asp to prevent association and subsequent dimerization, while preserving the net charge of the domain. The pH-dependent monomer–dimer equilibrium of NT$_{wt}$ can be monitored through the fluorescence shift of a single tryptophan (Trp) residue that becomes more exposed in the dimer[32]. The fluorescence ratio at 339/351 nm as a function of pH shows a sigmoidal relationship between monomer and dimer populations, with a pKa of dimerization at pH 6.5 (ref. 33). In contrast to NT$_{wt}$, the measured ratio for the novel mutant NT* corresponds to a monomer over the whole pH range (Fig. 2a). Size-exclusion chromatography (SEC) in the presence of 150 mM NaCl shows a clear correlation to the Trp fluorescence data measured under the same conditions. NT$_{wt}$ migrates as a monomer at pH 8 and a dimer at pH 5.5, with a 1.4-fold difference in hydrodynamic radius (Fig. 2c,d). The <2 difference in apparent size between the two states is predictable from the more compact structure of NT$_{wt}$ subunits in the dimer than in the monomer[32]. To further verify the correlation between Trp fluorescence and SEC data, we investigated several previously reported mutants and could confirm that a constitutive monomer mutant NT$_{A72R}$ (ref. 32) and a constitutive dimer mutant NT$_{E79QE84QE119Q}$ (ref. 33) are pH insensitive and migrate as the NT$_{wt}$ monomer and dimer, respectively (Fig. 2c,d). One particular mutant, NT$_{D40NE79QE119Q}$, adopts a conformation in between monomer and dimer based on Trp fluorescence data at high pH[33] and equivalently, it migrates with an intermediate hydrodynamic radius at pH 8, reflecting the

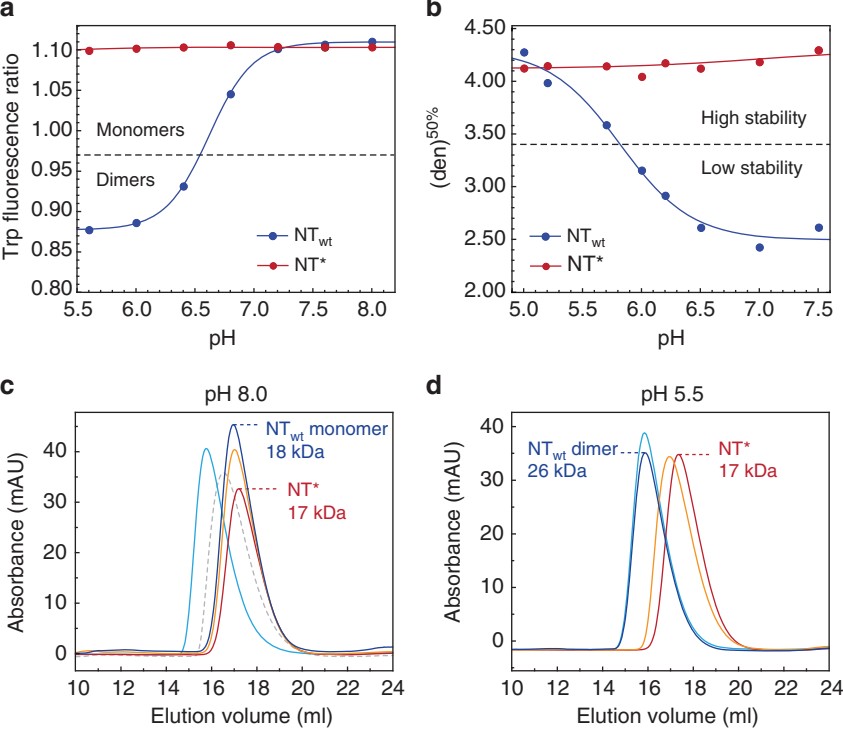

**Figure 2 | Characterization of the charge-reversed mutant NT\*.** (**a**) Monomer–dimer equilibrium measured with Trp fluorescence. Spectra between 300 and 400 nm were measured and the ratios at 339/351 nm (wavelengths corresponding to monomer/dimer conformations) were plotted as a function of pH for $NT_{wt}$ (blue) and NT\* (red). (**b**) Stability of $NT_{wt}$ (blue) and NT\* (red) in the presence of 0–7 M urea, measured with Trp fluorescence and presented as transition points between native and denatured states ($[den]^{50\%}$) as a function of pH. (**c**) SEC analysis at pH 8 shows the $NT_{wt}$ monomer (blue) with a hydrodynamic size similar to the constitutive monomer mutants $NT_{A72R}$ (orange) and NT\* (red). The migration profile for a dimer is represented by the constitutive dimer mutant $NT_{E79QE84QE119Q}$ (cyan). Another mutant, $NT_{D40NE79QE119Q}$ functions as a control and shows the profile expected in the presence of both monomers and dimers in equilibrium (dotted grey). (**d**) SEC analysis at pH 5.5 shows the $NT_{wt}$ dimer (blue) with a hydrodynamic size identical to the constitutive dimer mutant $NT_{E79QE84QE119Q}$ (cyan). At low pH, the constitutive monomer mutants $NT_{A72R}$ (orange) and NT\* (red) have similar migration profiles, comparable to those observed at pH 8, proving their inability to form dimers.

presence of both species (Fig. 2c). As expected from the Trp fluorescence spectra (Fig. 2a), the novel mutant NT\* migrates on SEC similar to the $NT_{wt}$ monomer in a pH-insensitive manner (Fig. 2c,d). In addition to these findings, 2D $^{15}N$–$^{1}H$ heteronuclear single quantum coherence (HSQC) NMR spectra for NT\* at both pH 7.2 and 5.5 are comparable to the spectrum measured for monomeric $NT_{wt}$ at pH 7.2 (Fig. 3a–c), demonstrating that the constitutively monomeric state of NT\* is recognized also at a structural level. The NMR spectra were recorded at pH 7.2 in the presence of 150 mM NaCl to reduce amide hydrogen exchange with water but at the same time give an $NT_{wt}$ monomer–dimer ratio comparable with that at pH 8 (ref. 33). When subjected to centrifugal filter concentration in this study, NT\* and $NT_{wt}$ could be concentrated to ~570 and ~310 mg ml$^{-1}$, respectively, before the proteins entered a gel state, indicating a higher intrinsic solubility for NT\*.

The protein stability at different pH was determined from urea-induced denaturation monitored with Trp fluorescence spectroscopy to evaluate the level of unfolding. As previously shown, $NT_{wt}$ is significantly more stable in the dimer conformation at low pH[33], while NT\* exhibits a pH-insensitive stability corresponding to the $NT_{wt}$ dimer (Fig. 2b). Comparable results were obtained using heat-induced denaturation measured with circular dichroism (CD) spectroscopy (Supplementary Fig. 1a). Notably, NT\* shows identical α-helical CD spectra before temperature induced unfolding and after refolding at pH 8 and 5.5 in contrast to $NT_{wt}$ that apparently has a lower refolding capacity, as judged from its about 15% lower CD amplitude at

205–225 nm after refolding, in particular at low pH (Supplementary Fig. 1b). Charge re-arrangement, as in NT\* compared to $NT_{wt}$, does not affect the total charge and we suggest that a less dipolar charge distribution together with a lower presence of destabilizing charge clusters in NT\* provides an explanation to the observed improved stability and refolding capacity.

**Comparison of different solubility tags.** Recombinant SP-C33Leu (rSP-C33Leu), KL4 (rKL4), rCCK-58 and rfhSP-D fused to $NT_{wt}$ and/or NT\* were in comparative experiments benchmarked against at least one additional fusion tag (PGB1, Trx or MBP) in terms of expression, solubility and fusion protein yield. The NT\* tag was thereafter removed to produce pure protein or peptide for further characterization. The other proteins and peptides included in this study (Aβ1-40, Aβ1-42, hCAP18, SP-C$_{ss}$ and β17) were expressed and purified as $NT_{wt}$ and/or NT\* fusion proteins and compared to published yields using other solubility tags (GST, SN, Ub, (NANP)$_{19}$ or IFABP) (see Supplementary Fig. 2 for sequences of all fusion constructs). To determine the solubility after expression in *E. coli* cells, samples were sonicated without the use of detergents and centrifuged to separate soluble and insoluble fractions. The $NT_{wt}$ and NT\* fusion proteins were all abundantly expressed at comparable levels, and exceeding those observed for other fusion tags used for comparative experiments (Supplementary Fig. 3a–d). It should be noted that the higher expression levels for NT are more

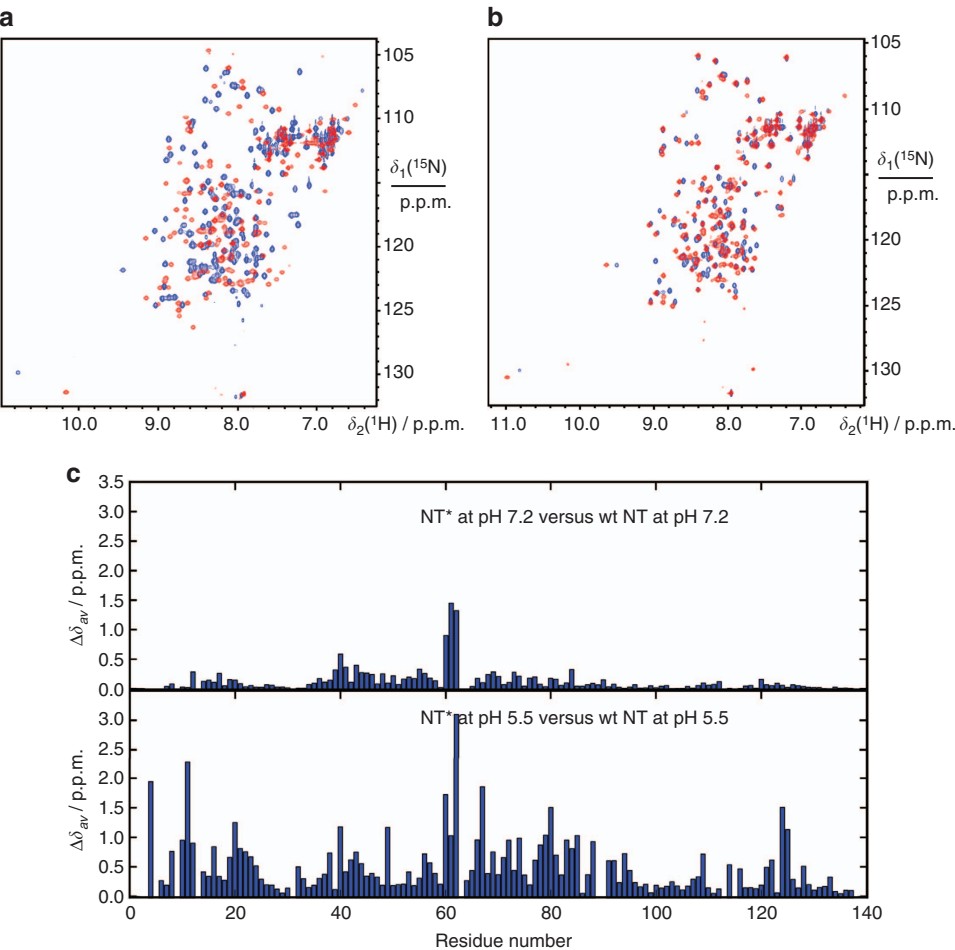

**Figure 3 | Comparison of NTwt and NT\* using 2D HSQC NMR.** (**a**) Overlay of $^{15}$N-$^{1}$H HSQC-NMR spectra of $NT_{wt}$ (red) and NT\* (blue) at pH 5.5. (**b**) Overlay of $^{15}$N-$^{1}$H HSQC-NMR spectra of $NT_{wt}$ (red) and NT\* (blue) at pH 7.2. (**c**) Averaged backbone amide $^{1}$H and $^{15}$N chemical shift differences $\Delta\delta_{av} = \sqrt{(0.1\Delta\delta_N)^2 + (\Delta\delta_H)^2}$ between $NT_{wt}$ and NT\* at pH 5.5 and pH 7.2.

pronounced than visually observed, since NT is less positively charged compared to the other tags and therefore becomes less stained by the Coomassie dye. NT\*, PGB1 and MBP were all potent mediators of solubility and the bulk of the fusion proteins were found in the soluble fractions (Fig. 4a–d). $NT_{wt}$ demonstrated a more divergent performance with the highest solubility in fusion with rfhSP-D (Fig. 4d) and ∼50% solubility in fusion with rKL4 (Fig. 4b). The most pronounced difference between NT\* and $NT_{wt}$ was seen in fusion with rSP-C33Leu, resulting in mainly insoluble fusion protein together with $NT_{wt}$, but mainly soluble fusion protein together with NT\* (Fig. 4a). Purification of NT\* fusion proteins on Ni-sepharose yielded 284, 428, 142 and 276 mg protein per litre culture for rSP-C33Leu, rKL4, rCCK-58 and rfhSP-D, respectively (Supplementary Fig. 4a–d and Supplementary Table 1). This corresponds to between two- and eightfold higher amounts than in fusion with PGB1, Trx or MBP, which can be mainly attributed to the higher expression levels (Supplementary Table 1). The yields using $NT_{wt}$ were intermediate, around 1.3- to 4-fold higher than in fusion with PGB1, while Trx was the least efficient solubility tag in fusion with rSP-C33Leu (Supplementary Fig. 4a). In addition to the comparative experiments, we also showed that NT\* can mediate solubility to recombinant amyloid β-peptides (rAβ1-40 and rAβ1-42, 4.3 and 4.5 kDa, respectively), hCAP18 (rhCAP-18, 16 kDa), β17 (rβ17, 7.4 kDa) and SP-C$_{ss}$ (rSP-C$_{ss}$, 3.6 kDa). All fusion proteins were abundantly expressed in a soluble form and they could be purified

at significantly higher yields compared to published yields using other tags (Supplementary Table 1).

**NT\*-TM peptides arrange into micelle-like particles.** SEC analysis of purified and soluble NT\*-rSP-C33Leu, with a calculated molecular mass of the monomer of 19 kDa, showed a well-defined oligomer population with an estimated size of 510 kDa, (Fig. 1c) corresponding to particles with a calculated hydrodynamic radius around 10 nm[64]. Transmission electron microscopy (TEM) of NT\*-rSP-C33Leu confirmed the presence of micelle-like particles with a size of 10–15 nm (Fig. 1d, Supplementary Fig. 5a) and particles of similar dimensions were also observed for NT\*-rKL4 (Supplementary Fig. 5b).

**A highly efficient purification procedure for TM peptides.** To optimize the downstream process, we developed a purification method independent of chromatographic steps for rSP-C33Leu and rKL4 expressed in fusion with NT\*. The non-chromatographic purification procedure is described in detail in Fig. 5. First, 1.2 M NaCl was added to the cleared bacterial lysate to precipitate the fusion proteins but leave most contaminating proteins in solution. Both fusion proteins were designed to have a methionine residue N terminal of the peptide, allowing for highly specific release of the Met-free rSP-C33Leu and rKL4 peptides with cyanogen bromide (CNBr) under acidic

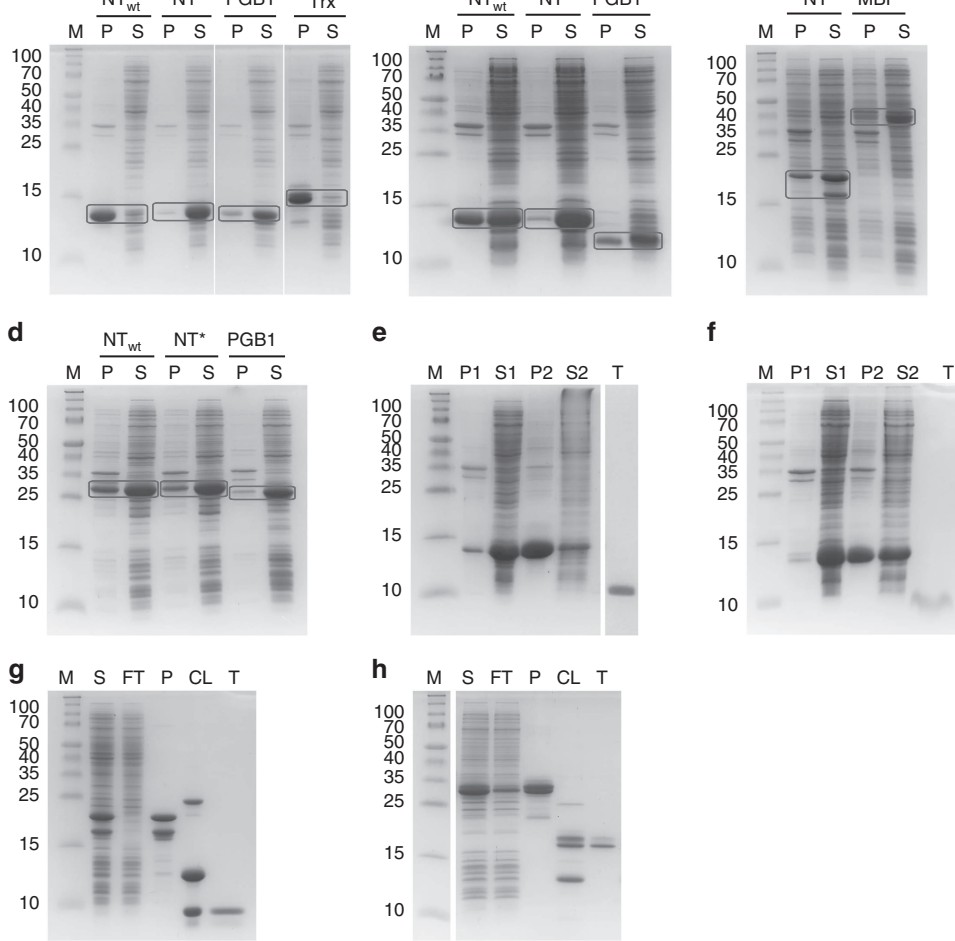

**Figure 4 | Solubility analysis of fusion proteins and subsequent purification of target peptides and protein.** Samples were analysed by SDS-PAGE and the molecular weights were compared to a protein standard (lane M). The molecular weights in kDa are given to the left of each gel figure. (**a–d**) Collected cells expressing peptides or protein in fusion with NT_wt, NT*, PGB1, Trx or MBP were sonicated and centrifuged to separate the soluble (S) and insoluble (P) fractions. Representative gels are shown for (**a**) rSP-C33Leu, (**b**) rKL4, (**c**) rCCK-58 and (**d**) rfhSP-D fusion proteins. For each fusion protein, the bands corresponding to the soluble and insoluble fractions are boxed. (**e–h**) NT* fusion proteins were further used for purification using different strategies. Surfactant peptides were purified by a simple NaCl precipitation/ethanol extraction protocol (Fig. 5) as shown for (**e**) rSP-C33Leu and (**f**) rKL4. The lanes represent insoluble fraction (P1), soluble fraction (S1), pellet after first NaCl precipitation (P2), supernatant after first NaCl precipitation (S2) and purified target peptide (T). Standard Ni-sepharose chromatography was used for purification of (**g**) rCCK-58 and (**h**) rfhSP-D. The lanes represent supernatant after sonication (S), column flow-through (FT), purified fusion protein (P), cleavage products with 3C protease (CL) and purified target protein (T).

conditions. Subsequent to CNBr cleavage, a second precipitation was performed using 0.8 M NaCl. Both rSP-C33Leu and rKL4 are soluble in organic solvents, for example, ethanol, methanol or isopropanol, and surprisingly all the NT-fragments generated by CNBr cleavage remained insoluble in these solvents, likely because the abundance of Met in NT (Supplementary Fig. 2) results only in small, polar fragments after CNBr cleavage. Accordingly, the precipitated pellet containing rSP-C33Leu could be further purified by suspension in 99.9% ethanol followed by centrifugation to isolate 20–30 mg pure rSP-C33Leu peptide per litre culture in the ethanol soluble fraction (Fig. 4e). The purification procedure was also applicable to rKL4, yielding 5–10 mg pure peptide per litre culture (Fig. 4f).

**Structural characterization of rSP-C33Leu.** NMR was used to characterize rSP-C33Leu in solution through the acquisition of homonuclear and heteronuclear 2D spectra. Sequence-specific [1]H assignments were obtained using standard procedures for small proteins[65] (see Supplementary Table 2 and Supplementary Fig. 6 for a complete list of proton chemical shifts and inter-residual NOE correlations, respectively). A structural model based on 20 conformers (Supplementary Fig. 7, Table 1) was built using the distance geometry software CYANA[66], where 100 randomly generated starting conformations were minimized against (i) NMR proton distance constraints derived from NOESY spectra and (ii) predicted dihedral angles, $\Psi$ and $\varphi$, obtained from proton, carbon and nitrogen chemical shifts[67]. According to the results, all conformers are characterized by a well-defined $\alpha$-helix, comprising residues 5–30, whereas peptide segments 1–4 and C terminal 31–33 are associated with flexible disordered regions. Ramachandran plot of dihedral angles $\Psi$ and $\varphi$ confirmed this interpretation exhibiting a typical $\alpha$-helix distribution for polypeptide segment 5–30.

A closer analysis of the rSP-C33Leu structure reveals that the length of the $\alpha$-helix, evaluated on the mean conformer and defined as the distance from the carbonyl carbon of Pro 5 to the amide nitrogen of Leu 30, is $\sim$37 Å. The length of the all-aliphatic part from the amide nitrogen of Leu 13 to the

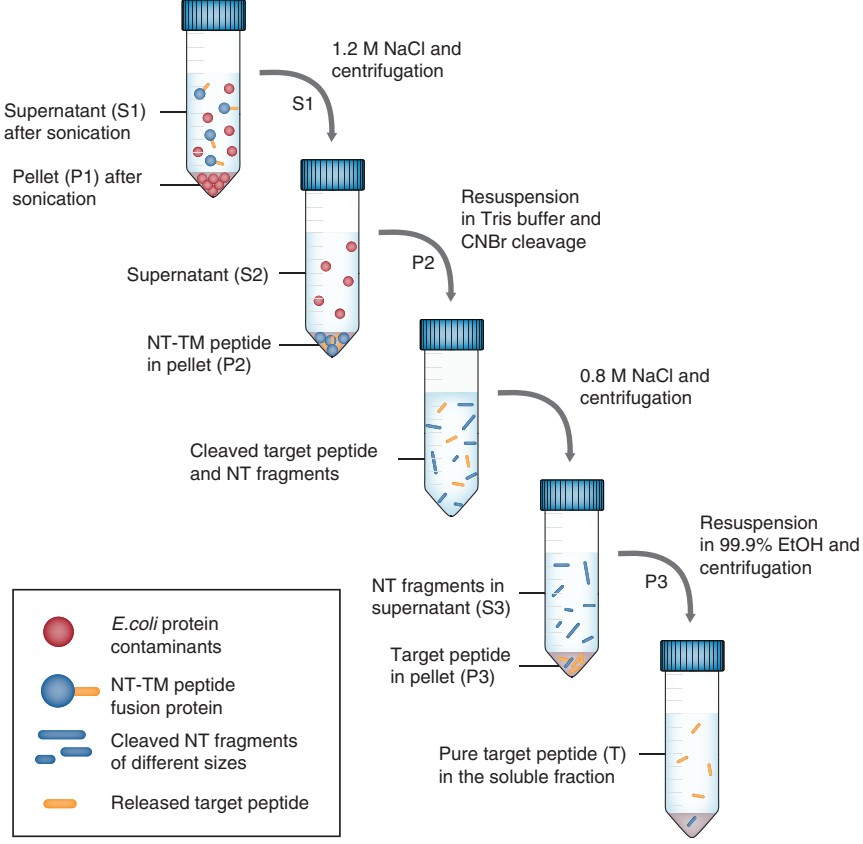

**Figure 5 | Schematic overview of the non-chromatographic purification of TM peptides.** The hydrophobic nature of surfactant peptides rSP-C33Leu and rKL4 allows for purification through salt precipitation and ethanol extraction. The figure describes each step in the purification protocol and further explains the denotations P1, S1, P2, S2 and T in Fig. 4e,f.

carbonyl carbon of Leu 26 is about 21 Å. Finally, the distances from the δ-methyl protons of a Leu residue (i) to those of Leu i + 2, which approximately correspond to the diameter of the helix in the central hydrophobic segment, are on average 10 Å. Comparison of the mean rSP-C33Leu conformer to the native SP-C structure[46] (PDB ID: 1SPF) determined in organic solvents, which well represents the SP-C structure in lipids[68], revealed a high degree of similarity (Fig. 6a) with 0.48 and 0.95 Å root-mean-square deviation (RMSD) values for backbone and heavy atoms, respectively, calculated for helix segment 9–30.

Ion mobility mass spectrometry (MS)[69] analysis of the gas-phase conformation of rSP-C33Leu was performed following electrospray ionization (ESI) from ethanol. The high-resolution mass spectrum of rSP-C33Leu revealed $[M + 2H]^{+2}$, $[M + 3H]^{+3}$ and $[M + 4H]^{+4}$ charge states at 1,798.51, 1199.17 and 899.60 monoisotopic $m/z$, respectively. The experimental monoisotopic MW is 3,594.64 ± 0.33 Da, in good agreement with the calculated monoisotopic MW 3,594.44 Da (average MW 3,596.74 Da). The $[M + 2H]^{+2}$, $[M + 3H]^{+3}$ and $[M + 4H]^{+4}$ ions were then assigned to their respective drift times upon ion mobility separation (Fig. 6b,c). Collisional cross sections (CCS) of 563.01 ± 0.31 Å$^2$ and 573.98 ± 1.29 Å$^2$ were determined for $[M + 3H]^{+3}$ and $[M + 2H]^{+2}$, respectively, suggesting similar compact conformations for these charge states. For $[M + 4H]^{+4}$ charge state, two mobility distributions of 572.55 ± 9.26 Å$^2$ and 655.87 ± 5.68 Å$^2$, respectively, were found, suggesting unfolding due to Coulombic repulsions. The CCS calculated from the rSP-C33Leu NMR structure is 647.63 Å$^2$, about 10% higher than the CCS obtained for the lower charge states, likely due to stronger intra-molecular interactions in the gas phase[70].

**The effect of rSP-C33Leu *in vitro* and in an *in vivo* model.** The effect of SP-C and analogues thereof on tidal volumes and lung-gas volumes (LGVs) can be evaluated *in vivo* in an animal model of RDS, using positive end-expiratory pressure (PEEP)[71]. Preterm newborn rabbits were treated at birth with 200 mg kg$^{-1}$ of preparations containing 2% rSP-C33Leu in a phospholipid mixture of dipalmitoylphosphatidylcholine (DPPC)/palmitoyloleoyl-phosphatidylglycerol (POPG) (68:31, w/w). Animals that received the same dose of phospholipids by treatment with Curosurf served as positive controls. Non-treated littermates were used as negative controls, since treatment with phospholipids only gives no improvement in tidal volumes or LGVs compared to non-treated controls[71]. The tidal volumes measured during ventilation were markedly increased for animals treated with 2% rSP-C33Leu in DPPC:POPG, compared to untreated negative controls, and were close to those obtained after treatment with Curosurf (Fig. 7a). At the end of the experiment, lungs were excised for LGV measurements using a water displacement technique[72,73]. The LGVs of animals treated with 2% rSP-C33Leu in DPPC:POPG were equal to those treated with Curosurf, and significantly higher than for non-treated animals (Fig. 7b). Likewise, the lung macroscopic appearances were similar for animals treated with 2% rSP-C33Leu in DPPC:POPG and Curosurf (Fig. 7c and Supplementary Fig. 8), as were the alveolar volume densities quantified by computer-aided image analysis[74] (Supplementary Fig. 9). These results are practically identical to those obtained using synthetic SP-C33 (ref. 51). Expression in a bacterial system and purification by a salt precipitation/ethanol extraction protocol could potentially lead to lipopolysaccharide (LPS) contamination. However, similar animal experiments were performed for up to 4 h without observing any

**Table 1 | Quantitative characterization of the 20 CYANA conformers used to represent the solution structure of rSP-C33Leu after energy minimization with the program OPAL.**

| Quantity | Value |
|---|---|
| *NMR constraints* | |
| Distance constraints | |
| Total NOE | 266 |
| Intra-residue | 162 |
| Inter-residue | 104 |
| Sequential ($|i-j|=1$) | 55 |
| Medium-range ($|i-j|\leq4$) | 49 |
| Long-range ($|i-j|\geq5$) | — |
| Intermolecular | — |
| Hydrogen bonds | 18 |
| Total dihedral angle restraints | |
| phi | 28 |
| psi | 28 |
| | |
| *Structure statistics* | |
| Violations* | |
| Distance constraints (Å) $>0.1$ Å | 0.05 ± 0.22 (0,…1) |
| Dihedral angle constraints (°) $>2.5°$ | 0 |
| Max. dihedral angle violation (°) | 0.70 ± 0.24 (0.53,…,1.69) |
| Max. distance constraint violation (Å) | 0.09 ± 0 (0.09,…,0.10) |
| Deviations from idealized geometry | |
| M/c bond lengths (Å) $>0.05$ Å (%) | 0 |
| M/c bond angles (°) $>10°$ (%) | 4.4 |
| Impropers (°) | — |
| RMSDs*,† (Å) | |
| Backbone of residues 5–30 | 0.44 ± 0.19 (0.20,…,0.95) |
| All heavy atoms of residues 5–30 | 0.91 ± 0.14 (0.72,…,1.23) |

*Mean ± s.d. (range).
†RMSD values are calculated on 20 structures with respect to the mean structure in the residue range 5–30, where an α-helix secondary structure is expected according to the backbone hydrogen bonds electrostatic criteria[79] and Ramachandran dihedral angle distributions.

adverse reactions (Supplementary Table 3), supporting that the rSP-C33Leu preparations contain low amounts of LPS.

We further investigated the *in vitro* surface activity of rSP-C33Leu in a captive bubble surfactometer (CBS)[75]. The surface tension and compressibility was measured for 2% rSP-C33Leu reconstituted in DPPC:POPG (68:31, w/w) and compared to Curosurf. The median values from three experiments revealed a higher maximum surface tension and degree of compression required to reach $5\,mN\,m^{-1}$ for vesicles containing 2% rSP-C33Leu in DPPC:POPG (Supplementary Table 4). The slightly higher surface activity of Curosurf is in accordance with *in vivo* experiments and can be attributed to its more complex phospholipid composition and presence of both of the hydrophobic-surfactant proteins SP-C and SP-B[76].

**Recombinant production of rfhSP-D and rCCK-58.** rfhSP-D and rCCK-58 in fusion with NT* were designed to contain a recognition sequence for coxsackievirus 3C protease just N terminal of the target proteins to allow site-specific cleavage under mildly reducing conditions. Purification of the fusion proteins on Ni-sepharose, cleavage with 3C protease, and a second round of purification to remove the tag yielded 16 mg rCCK-58 peptide per litre culture (Fig. 4g) and 85 mg rfhSP-D protein per litre culture. (Fig. 4h) and there is still a potential to increase the recovery in several of the purification steps. The 6.8 kDa rCCK-58 and 18.7 kDa rfhSP-D migrated as expected on SDS-PAGE (Fig. 4g,h). For rfhSP-D, an upper band of lower intensity could be observed in addition to the major band (Fig. 4h). When subjected to SEC, the majority of rfhSP-D eluted as a single population corresponding to 100 kDa according to a set of calibrants (Fig. 8a). The main eluting peak was isolated and appeared

similar to the non-separated sample when analysed on SDS-PAGE (Fig. 8b), indicating that the observed bands represent two SDS-stable conformations that migrate with the same hydrodynamic size using SEC. Since the non-uniform fold of rfhSP-D, comprising globular, coil-coiled and extended regions, may lead to an over-estimation of the molecular mass using SEC, we further analysed the protein using ESI-MS. This confirmed that the main part of rfhSP-D adopts the 57 kDa trimer conformation that is essential for activity (Fig. 8c and Supplementary Fig. 10).

**Discussion**
Inspired by how spiders store their aggregation-prone silk proteins at extremely high concentration, we have developed a novel method to produce aggregation-prone peptides and proteins in heterologous hosts. A designed mutant of a spider silk protein domain, NT*, is unable to dimerize and displays markedly increased solubility, stability and refolding capacity compared to NT$_{wt}$. In a comparative study, we showed that NT* allows for soluble expression of the TM peptides rSP-C33Leu and rKL4 as well as of the surfactant protein fragment rfhSP-D and the cholecystokinin peptide rCCK-58, all which previously have been reluctant to recombinant production in *E. coli*. Purification of NT* fusion proteins yielded up to eightfold higher amounts compared to PGB1, Trx and MBP fusion proteins, and all peptides/protein were produced in a soluble form after removal of the tag. We also showed that the NT* solubility tag can be used to produce several other recombinant proteins and peptides of biomedical relevance, including rAβ1-40, rAβ1-42, rhCAP-18, rβ17 and rSP-C$_{ss}$, with fusion protein yields well exceeding those previously published using other solubility tags (Supplementary Table 1).

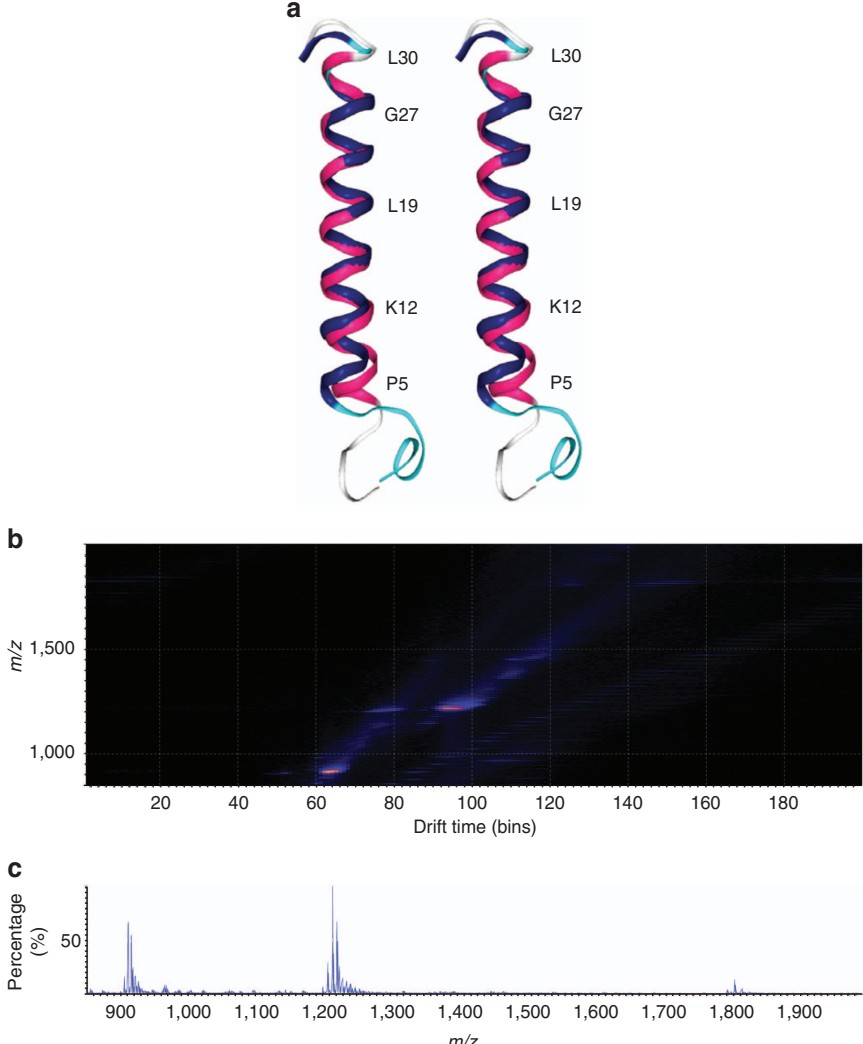

**Figure 6 | Structural characterization of rSP-C33Leu.** (**a**) Stereo view showing an overlay of the mean conformer representing the three-dimensional structure of rSP-C33Leu (magenta) with the native porcine SP-C structure (blue) (PDB ID: 1SPF). (**b**) Mobility map of rSP-C33Leu in ethanol. (**c**) ESI mass spectrum showing the $[M+2H]^{+2}$, $[M+3H]^{+3}$ and $[M+4H]^{+4}$ charge state envelopes.

We hypothesized that the amphipathic nature of NT*-TM peptide fusion proteins would allow them to arrange into micelle-like particles, thus protecting the water-insoluble peptide during expression and purification in aqueous solvents. This hypothesis was verified from SEC and TEM analysis of purified NT*-rSP-C33Leu and NT*-rKL4, showing a homogeneous population of ~10 nm particles. In addition, NT* also functioned as a general solubility tag for water-soluble proteins that are prone to misfold or aggregate during heterologous expression. The underlying mechanism for solubilizing hydrophilic peptides or proteins remain to be established, but it is likely mediated by the remarkably high inherent solubility and folding capacity of NT*, rather than formation of micelle-like particles.

To optimize the downstream processes, we developed a method to obtain pure TM peptides without the use of chromatography. NT* allows for efficient purification of hydrophobic target peptides using just simple NaCl precipitation and ethanol extraction step since none of the CNBr-cleaved fragments of NT* dissolve in the ethanol fraction. The procedure described herein is amenable to scale-up and represents a cheap, efficient and, from a regulatory point of view, beneficial way of producing non-animal derived rSP-C33Leu and rKL4 for future clinical use.

rSP-C33Leu produced with this method has correct covalent structure, is structurally very similar to the native SP-C peptide as judged by NMR spectroscopy, and a mixture of rSP-C33Leu and synthetic phospholipids has therapeutic effects in an animal model of RDS, similar to the porcine-derived surfactant Curosurf. Current therapeutic surfactants contain the hydrophobic-surfactant components, required to restore basal lung function but lack the hydrophilic constituents SP-A and SP-D. Much effort has been made to develop recombinant versions of SP-A and SP-D, but rfhSP-D, previously accumulated into inclusion bodies during expression in *E. coli* and, consequently, the production was hampered by the requirement of denaturing agents and subsequent refolding. In this paper, we now show that rfhSP-D can be expressed in a soluble form in *E. coli* when fused to the NT* solubility tag, and the protein fragment is able to adopt a trimeric conformation that is indicative of native folding, which allows further investigation of its functions and the therapeutic potential in models of human respiratory disease.

In summary, we herein present a novel solubility enhancing fusion tag that allows bacterial expression of a panel of pharmaceutically relevant peptides and proteins with different biochemical properties that previously have been refractory to

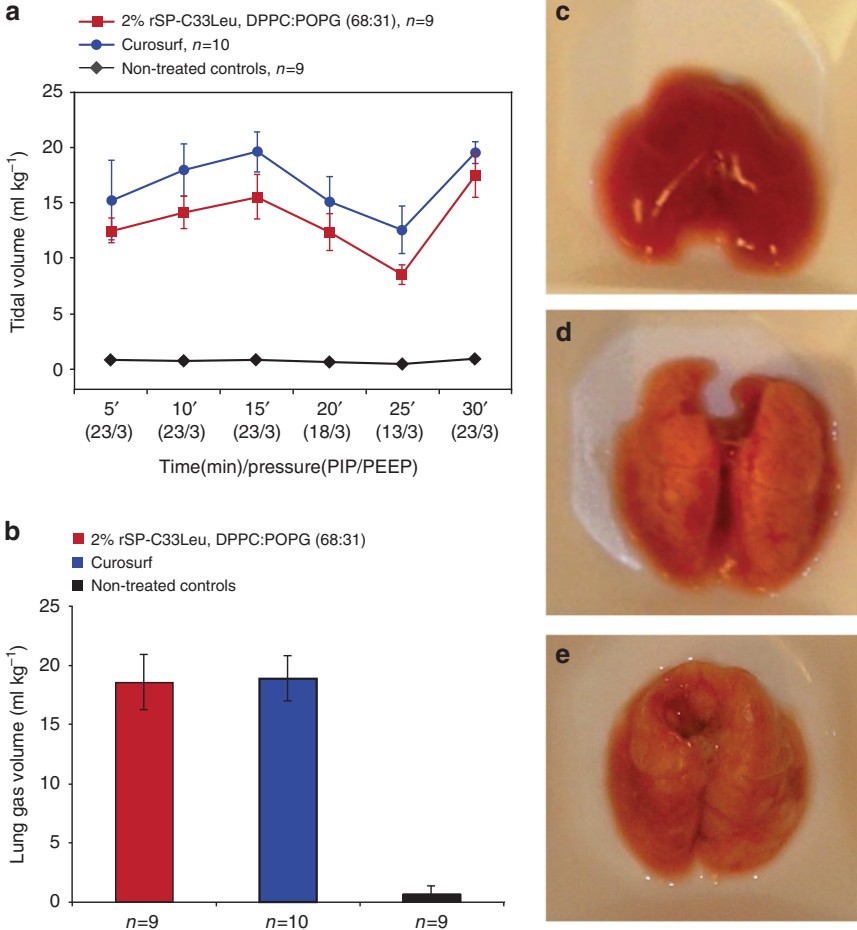

**Figure 7 | Effects of rSP-C33Leu in animal model of RDS.** Immature newborn rabbits were treated at birth with 200 mg kg$^{-1}$ of 2% rSP-C33Leu in DPPC:POPG (68:31, w/w) and compared to animals receiving the same dose of Curosurf as positive and non-treated animals as negative controls. The results are presented as median values ± M.A.D. (median absolute deviation) as indicated by error bars, and n is the number of animals. (**a**) Tidal volumes were measured during 30 min of ventilation with different peak inspiratory pressures (PIP) and constant positive end-expiratory pressure (PEEP). Treatment with 2% rSP-C33Leu in DPPC:POPG or Curosurf showed similar results, with significantly increased tidal volumes compared to non-treated animals (Newman–Keuls test, $P < 0.0005$) (**b**) The LGVs of animals treated with 2% rSP-C33Leu in DPPC:POPG were equal to those of animals treated with Curosurf, and significantly higher than those for non-treated animals (Newman–Keuls test, $P < 0.0005$). (**c**–**e**) Lung appearances at the end of the experiment are shown as representative photographs of whole lungs with median LGV for (**c**) non-treated control animals (LGV: 0.7 ml kg$^{-1}$) and animals treated with (**d**) 2% rSP-C33Leu in DPPC:POPG (68:31) (LGV: 18.6 ml kg$^{-1}$) or (**e**) Curosurf (LGV: 18.9 ml kg$^{-1}$). Appearances of the whole set of analysed lungs are shown in Supplementary Fig. 8.

recombinant production. We benchmark the performance of NT* to several commonly used tags and conclude that fusion to NT* gives up to eight times higher protein yields. NT* also allows TM peptide purification to homogeneity without the use of chromatography, and the production of functional synthetic lung surfactant preparations.

## Methods

**Site-directed mutagenesis.** The previously described vector pT7HisTrxHisNT[36], containing the *E. australis* MaSp1 NT$_{wt}$ sequence, was subjected to consecutive point mutations of D40K followed by K65D using the QuickChange site-directed mutagenesis kit (Agilent Technologies, Santa Clara, CA, USA) according to the manufacturer's recommendations. After sequence verification, the plasmid was digested with restriction enzymes EcoRI and HindIII to isolate the DNA fragment containing the mutated NT* sequence. The fragment was purified on a 2% agarose gel and ligated into pT7HisTrxHisNT, previously digested with the same enzymes and purified on 1% agarose gel to remove NT$_{wt}$. The ligation mixture was heat-shock transformed into chemically competent *E. coli* Nova Blue cells followed by plasmid preparation and sequence verification.

**Expression and purification of NT$_{wt}$ and NT*.** The plasmids pT7HisTrxHisNT$_{wt}$ and pT7HisTrxHisNT* were transformed into chemically competent *E. coli* BL21

(DE3) cells. Colonies were inoculated to 10 ml Luria–Bertani (LB) medium with 70 mg l$^{-1}$ kanamycin and grown at 37 °C and 180 r.p.m. overnight. Overall, 5 ml overnight culture was inoculated to 500 ml LB medium (1/100) with kanamycin and cells were further grown at 37 °C to OD$_{600}$ ∼1. Expression was induced by addition of isopropyl β-D-1-thiogalactopyranoside (IPTG) to a final concentration of 0.5 mM and the culture was further incubated at 30 °C, 180 r.p.m. for 4 h. Cells were harvested by centrifugation, resuspended in 20 mM Tris, pH 8 to 30 ml and stored at −20 °C for at least 24 h. Cell lysis was performed on ice for 1 h in the presence of lysozyme (1 mg ml$^{-1}$), DNAse (1 μg ml$^{-1}$) and MgCl$_2$ (2 mM). The supernatant was cleared by centrifugation at 27,000g for 30 min. Proteins were purified on Immobilized Metal Ion Affinity Chromatography (IMAC) columns previously packed with Ni-Sepharose (GE Healthcare) and equilibrated with loading buffer (20 mM Tris, pH 8). Bound protein was washed with 20 mM Tris, 5 mM imidazole, pH 8 and eluted with 20 mM Tris, 300 mM imidazole, pH 8 in 1 ml fractions. The absorbance at 280 nm was measured for each fraction, and protein-containing fractions were pooled. Imidazole was removed by over-night dialysis at 4 °C and in 5 l loading buffer, using a Spectra/Por dialysis membrane with a 6–8 kDa molecular weight cut-off. Dialysis was performed in the presence of 1/1,000 w/w thrombin to proteolytically release the fusion tag. The cleaved and dialyzed sample was loaded to Ni-Sepharose to bind the His-Trx tag, and unbound target protein was collected. The purity of the protein in each step was determined by SDS-PAGE using a 15% acrylamide gel stained with Coomassie Brilliant Blue. For expression of $^{15}$N-labelled NT$_{wt}$ and NT* for HSQC-NMR analysis, the same procedure was used except that M9 minimal medium containing $^{15}$NH$_4$Cl as the sole nitrogen source was used.

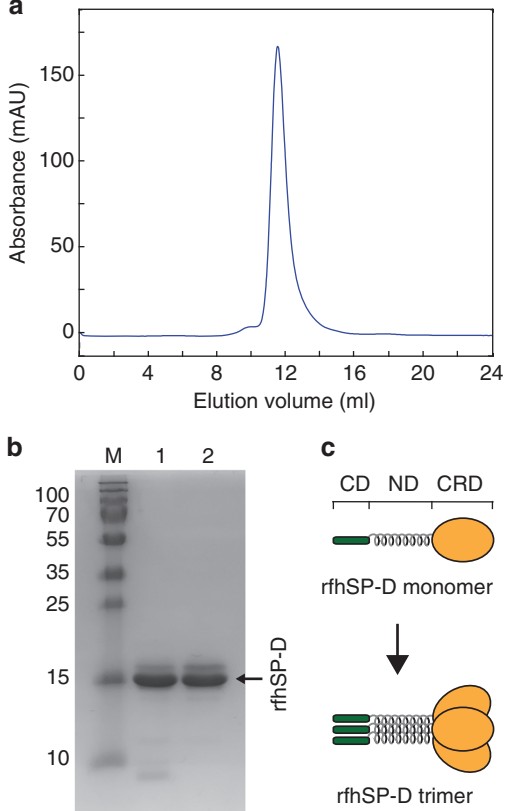

**Figure 8 | rfhSP-D adopts a trimer conformation.** (**a**) Size-exclusion chromatogram of purified rfhSP-D, migrating as a well-defined population of trimeric protein, as confirmed with ESI-MS (Supplementary Fig. 10). (**b**) SDS-PAGE comparison of rfhSP-D after Ni-sepharose purification (lane 1) and after SEC separation of the main eluting peak (lane 2) shows an unchanged distribution of SDS-stable conformations. The molecular weights in kDa of a protein standard (lane M) are given to the left of the gel figure. (**c**) The rfhSP-D monomer comprises eight Gly-Xaa-Yaa repeats from the collagenous domain (CD), the α-helical neck domain (ND) and the carbohydrate-recognition domain (CRD). The functional trimer is formed as the neck domains from three monomer units assemble into a coiled-coil motif.

**Centrifugal filter concentration.** Purified $NT_{wt}$ and NT* proteins were concentrated by ultrafiltration in two steps to determine the concentration limit. For each protein, around 20 mg at a concentration of 2 mg ml$^{-1}$ was pre-concentrated by centrifugation at 4,700$g$ and 4 °C in a Vivaspin 20 centrifugal tube with a 5 kDa molecular weight cut-off (GE Healthcare) until a volume of 1 ml was reached. The concentration was continued by centrifugation at 15,000$g$ and 4 °C in a Vivaspin 500 centrifugal tube with a 3 kDa molecular weight cut-off (GE Healthcare). The final protein concentrations were determined from the absorbance at 280 nm of samples taken just before the proteins entered a gel state.

**Tryptophan fluorescence measurement.** Fluorescence emission spectra were measured on a spectrofluorometer (Tecan Safire 2) using Costar black polystyrene assay plates with 96 flat bottom wells. The proteins were diluted to a concentration of 10 μM in 20 mM HEPES/20 mM MES adjusted to pH 5.6–8 in steps of 0.4 pH units. After exciting the samples at 280 nm (5 nm bandwidth), emission spectra were recorded in 1 nm steps between 300 and 400 nm (10 nm bandwidth). The tryptophan fluorescence ratio was calculated from the fluorescence intensities at 339 nm and 351 nm, and plotted as a function of pH. The data obtained for $NT_{wt}$, was fitted to a two-state binding model due to the sigmoidal behaviour of the monomer–dimer equilibrium.

**Size-exclusion chromatography of $NT_{wt}$ and mutants.** $NT_{wt}$, NT* and previously reported mutants $NT_{A72R}$, $NT_{E79QE84QE119Q}$ and $NT_{D40NE79QE119Q}$ (ref. 33), were purified according to protocol and analysed by SEC. The proteins were diluted to 2 mg ml$^{-1}$ in running buffer (either 20 mM Tris, 150 mM NaCl, 1 mM EDTA, pH 8 or 20 mM MES, 150 mM NaCl, 1 mM EDTA, pH 5.5) and

incubated at room temperature for 30 min before analysis. A Superdex 75 column was equilibrated in running buffer and 100 μl samples were run through the column at a rate of 0.5 ml min$^{-1}$. Elution of protein was detected by measuring optical absorbance at 280 nm. Molecular weight standards conalbumin (75 kDa), ovalbumin (43 kDa), carbonic anhydrase (29 kDa) and ribonuclease A (13.7 kDa) (GE Healthcare) were run as above. Shown in the same order, the elution volumes were 12.56, 13.68, 15.43 and 18 ml, respectively, at pH 8 or 12.58, 13.85, 15.41 and 18.18 ml, respectively, at pH 5.5.

**HSQC-NMR measurements.** $^{15}$N-labelled samples for comparison of $NT_{wt}$ and NT* were prepared in either 20 mM sodium phosphate, 20 mM NaCl, pH 5.5 or 20 mM sodium phosphate, 300 mM NaCl, pH 7.2 buffers. The 2D $^{15}$N–$^1$H HSQC-NMR spectra were recorded at 25 °C on a Varian Unity Inova 600-MHz NMR spectrometer equipped with an HCN cold probe. Assignment of the backbone amide group resonances of NT* was obtained on the basis of $NT_{wt}$ assignments at pH 7.2 (ref. 32) by analysing a 3D $^{15}$N-resolved NOESY-HSQC spectrum acquired with a 60 ms mixing time. The spectra were processed using Topspin 3.1 (Bruker) and analysed in CARA[77] (freeware).

**Urea-denaturation.** Protein was diluted to 5 μM in 20 mM HEPES/20 mM MES supplemented with 0–7 M urea in 0.5 M steps. The stability of the protein at each concentration of urea was monitored with Trp fluorescence at constant pH values ranging from 5 to 7.5 with 0.5 unit steps. For each measured pH, the fluorescence ratio was plotted against the urea concentration and fitted to a two-state unfolding model to determine the transition points. The data was then presented as transition points between native and denatured states ($[den]^{50\%}$) as a function of pH.

**CD spectroscopy.** Experiments were performed on a 410-model CD spectrometer (Aviv biomedical, Lakewood, NJ, USA) using 300 μl cuvettes with a 1 mm path length. For all measurements, the proteins were diluted to 10 μM in 5 mM phosphate buffer at pH 5.5 or pH 8. Spectra were recorded from 260 to 185 nm at 25 °C, after heating to 95 °C and again at 25 °C after the samples were allowed to cool down. For each temperature, the data is shown as an average of four scans. Temperature scans were measured at 222 nm by recording 1 °C steps in the temperature interval 25–95 °C. The CD signal was converted to fraction folded according to the formula $([CD]_{obs} - [CD]_D)/([CD]_N - [CD]_D)$, where CD is the signal measured in millidegrees, $CD_D$ is the signal for the denatured state, $CD_N$ is the signal for the native state and CDobs is the signal at each data point in between N and D. The data were plotted as a function of temperature (°C) and fitted to a two-state unfolding model to obtain the melting temperatures ($T_m$) at the equilibration points.

**Expression of fusion proteins.** Constructs containing a His$_6$ tag, followed by a solubility tag ($NT_{wt}$, NT*, PGB1, MBP or Trx), a cleavage site (3C protease, tobacco etch virus (TEV) protease, thrombin or CNBr) and a target protein/peptide (rSP-C33Leu, rKL4, rCCK-58, rfhSP-D, rAβ1-40, rAβ1-42, rhCAP-18, rβ17 or rSP-C$_{ss}$) were cloned into pT7 vectors (see Supplementary Fig. 2 for sequences) and subsequently transformed into chemically competent *E. coli* BL21 (DE3) cells or Origami 2 (DE3) cells (only for NT*-rhCAP-18). Plasmid-containing cells were inoculated to 10 ml LB medium with 70 mg l$^{-1}$ kanamycin and grown at 37 °C and 180 r.p.m. overnight. Overall, 5 ml over-night culture was inoculated to 500 ml LB medium (1/100) with kanamycin and cells were further grown at 30 °C to $OD_{600} \sim 1$. The cells were induced by addition of IPTG to a final concentration of 0.5 mM and expression was performed at 20 °C overnight. The day after, cells were collected by centrifugation, resuspended in 20 mM Tris, pH 8 to 30 ml and stored at −20 °C for at least 24 h. Comparable amounts of cells were taken before induction and after expression and analysed by SDS-PAGE using 15% acrylamide gels stained with Coomassie Brilliant Blue.

**Purification of fusion proteins for comparison of yields.** All of the investigated fusion proteins were first solubilized and purified using similar protocols to directly compare the fusion protein yields. Protocols for further purification of hydrophobic TM peptides (rSP-C33Leu and rKL4) and hydrophilic protein/peptide (rfhSP-D and rCCK-58) are described in separate sections below. Fusion proteins were solubilized by sonication at 80% amplitude, 1 s on and 1 s off for a total of 3 min for constructs containing TM peptides and typically 2 min for all other proteins. NT*-rhCAP-18 and NT*-rAβ1-40/rAβ1-42 were fully solubilized by sonication in the presence of 2 and 8 M urea, respectively. The soluble and insoluble fractions were separated by centrifugation at 27,000$g$, 4 °C for 30 min. The clear lysates were purified on Ni-Sepharose IMAC columns and dialyzed as described above but without cleaving the fusion protein. Comparable samples were taken from pellets and supernatants after sonication and from the purified fusion proteins for SDS-PAGE analysis using 15% acrylamide gels stained with Coomassie Brilliant blue.

**Size-exclusion chromatography of NT*-rSP-C33Leu.** Purified fusion protein was diluted to 2 mg ml$^{-1}$ in running buffer (20 mM Tris, 150 mM NaCl, 1 mM EDTA,

pH 8). A Superdex 200 column was equilibrated in running buffer and 200 µl of the sample was run through the column at a rate of 0.5 ml min$^{-1}$. Elution of protein was detected by measuring optical absorbance at 280 nm. Molecular weight standards ferritin (440 kDa), aldolase (158 kDa), conalbumin (75 kDa), ovalbumin (43 kDa), carbonic anhydrase (29 kDa) and ribonuclease A (13.7 kDa) (GE Healthcare) were run and eluted at 8.56, 10.65, 12.06, 12.96, 14.26 and 15.64 ml, respectively.

**Transmission electron microscopy.** The samples were diluted in 20 mM Tris, pH 8. For negative staining, 3 µl samples were applied to glow-discharged carbon-coated copper grids, stained with 2% (w/v) uranyl acetate and air-dried. The grids were checked using JEOL JEM-2100f transmission electron microscope operated at 200 kV. Images were collected with TVIPS TemCam-F415 4k × 4k CCD-camera (Tietz Video and Image Processing Systems GmbH, Gauting, Germany) using a nominal magnification of × 60,000.

**Purification of TM peptides by precipitation and extraction.** Cells were lysed by sonication at 80% amplitude, 1 s on and 1 s off, for 3 min in total time. The sonication procedure was repeated once more after standing on ice for 5 min and the sample was centrifuged at 50,000g for 30 min. Sodium chloride was added to the supernatant to a final concentration of 1.2 M and the centrifugation was repeated. The pellet from centrifugation was dissolved in 20 mM Tris, pH 8 and sonicated at 60% amplitude, 1 s on and 1 s off, for 3 min in total time to fully re-dissolve the fusion protein. CNBr cleavage was performed at pH 1 by adding 1.7 ml 2 M HCl to 30 ml dissolved solution, followed by 1.7 ml 1 M CNBr. The cleavage reaction was performed overnight at room temperature. The next day, 800 mM sodium chloride was added to the cleavage reaction in a second precipitation step, followed by centrifugation at 20,000g for 30 min. The supernatant was removed and the pellet was dried at 37 °C and suspended in 99.9% ethanol. Insoluble material was removed by centrifugation at 20,000g for 30 min.

**NMR spectroscopy.** Preliminary tests were conducted to assess the solubility of rSP-C33Leu in a variety of pure non-polar solvents including CD$_3$OD, CDCl$_3$ and a 150 mM mixture of SDS in deuterated water. Satisfactory results were achieved using a mixture formed by CDCl$_3$/CD$_3$OD/0.1 M HCl 32:64:5 (v/v), which allowed full solubilization of the sample. Approximately 25 mg of the rSP-C33Leu dry peptide was dissolved in a CDCl$_3$/CD$_3$OD 33:66 (v/v) solution to obtain a final compound concentration of ~1.7 mM. On complete dissolution, an aliquot of 816 µl was supplemented with 42 µl of a 0.1 M HCl solution prior to be transfered into a standard 5 mm NMR tube. The analysis was performed on a Bruker AVANCE III HD 600 spectrometer operating at the proton resonance frequency of 600 MHz equipped with a 5 mm TCI inverse triple resonance cryoprobe H-C/N-D-0.5-Z ATMA. COSY, TOCSY and NOESY (200 ms mixing time) spectra were acquired at 25 °C with a solvent presaturation module –238 mW hard pulse applied for 2 s—in States-TPPI pure phase absorption mode.

**Spin system identification and sequence-specific assignment.** Close inspection of the Hα-HN fingerprint region of COSY and TOCSY spectra revealed the presence of 32 peaks. Since in the peptide primary structure there are two glycines—which would give rise to a double signal due the presence of two α protons in each residue—and two prolines—which would not be present in the fingerprint region because of the lack of NH groups—it was possible to count 30 non-prolinic residues out of 31. Different spin systems were finally assigned to specific amino acid types following the general schemes for small proteins[65]. The NOESY inter-residual correlation peaks were analysed to link the identified spin systems to specific positions in the primary sequence. The analysis started from those residues that could more easily be identified such as valine, histidine, arginine and alanine residues which appear only once in the peptide structure. In particular, they are found in position 6, 7, 10 and 28, respectively. The identification of the position of the remaining spin systems was done by looking at the NOESY correlation patterns of type $H\alpha_i - NH_{i+1}$, $NH - NH$ and $H\alpha_i - NH_{i+3}$.

**De-novo structure modelling.** The computational approach was performed using the distance geometry program CYANA (L.A. Systems) with 100 randomly generated starting conformations that were subjected to minimization against the NMR input data including: (i) dihedral angles Ψ and φ and (ii) upper distance restraints obtained through integration and calibration of NOE peaks. Dihedral angular constraints were derived from proton and heteronuclear carbon and nitrogen HSQC spectra using the TALOS+ server[67], a freeware Java-based platform performing the prediction of protein backbone torsion angles from the NMR chemical shifts. According to the results, 56 Ψ and φ dihedral angles were defined by the programme with an acceptable level of confidence. Upper distance restraints were obtained through calibration of NOE peaks volumes using a built-in CYANA macro. A total of 266 distance bounds were identified. Among those, 162 were associated to intra-residue distances whereas the remaining ones were attributed to inter-residue NOE connectivities. The 20 best conformers with the smallest target function were energy-minimized *in vacuo* using a modification of the AMBER force field implemented in OPAL[78] (part of CYANA software,

L.A Systems). Table 1 represents a survey of the parameters which afford a quantitative evaluation of the quality of the structure determination. For all structures, the energy refinement procedure allows the stabilization of the conformers by reducing the quality of energy associated (that is, − 731 versus. − 1,117 kcal mol$^{-1}$) by almost 35%. Significant improvements were also registered for the number of distance upper bounds violations, which decreased on average from 5.60 to 0.05. Analysis of the RMSDs calculated between the 20 refined peptide structures and the mean conformer were all within the range of acceptance exhibiting values of 0.44 and 0.91 Å for backbone and heavy atoms, respectively.

**Ion mobility mass spectrometry.** An aliquot of 1 mg of rSP-C33Leu was dissolved in 1 ml of ethanol and the stock solution was then further diluted to 0.2 mg ml$^{-1}$ in ethanol. The sample was directly infused into the ion source at 5 µl min$^{-1}$ and all experiments were performed in duplicate. MS measurements were performed on a Waters SynaptG2S Q-TOF equipped with a standard ESI source. The following ionization parameters were used: polarity, ES +; capillary, 30 kV; source temperature, 100 °C; sampling cone, 80; source offset, 80; source gas flow, 0 ml min$^{-1}$; desolvation temperature, 300 °C; cone gas flow, 0 l h$^{-1}$; desolvation gas flow, 600 l h$^{-1}$; nebulizer gas flow, 6 bar. When operating in time of flight (TOF) mode, the instrument was used in high-resolution set-up and mass spectra were acquired for 0.5 min at 0.5 s scan time starting from $m/z$ 500–2,000. High-resolution mass spectrometric calibration was performed infusing a standard solution of sodium iodide at 5 µl min$^{-1}$ in between the $m/z$ range 100–2,000. Calibration was accepted when RMS was below 1 p.p.m. The same parameters were adopted when the instrument was operated in mobility TOF mode. Data were acquired and elaborated with MassLynxv4.1 software (Waters). For ion mobility measurement, the IMS travelling wave velocity (T-Wave) was set at 1,000 m s$^{-1}$ and IMS T-wave height was set at 40 V. Optimization of Tri-wave mobility parameters and the evaluation of the impact on the protein structure was performed. Mobility calibration was performed using a freshly prepared solution of poly-DL-alanine at 1 mg ml$^{-1}$ in water/acetonitrile 50/50, which was directly infused into the ion source at 5 µl min$^{-1}$. The mobility of poly-DL-alanine clusters was evaluated under the optimal ion mobility parameters used for the investigation of the protein of interest. A literature reference table containing CCS values in Å$^2$ was used to calibrate the acquired poly-DL-alanine spectrum. Driftscope 2.7 software tool (Waters) was used to elaborate ion mobility data, including the application of mobility calibration. Peak detection was performed setting a minimum drift peak width (FWHM) at 2.0 bins and an MS resolution of 1,000 to centroid on average mass. Following peak annotation based on the high-resolution isotopic distribution, collisional cross section (CCS, Ω) values were charge adjusted.

**In vivo experiments.** Surfactant preparations were tested in preterm newborn rabbits obtained at a gestational age of 27 days (term 31 days). The animals were tracheotomized at birth and received via the tracheal cannula 2.5 ml kg$^{-1}$ of synthetic preparations containing 2% rSP-C33Leu in dipalmitoylpho-sphatidylcholine (DPPC)/palmitoyloleoyl-phosphatidylglycerol (POPG) 68:31 (w/w) at a concentration of 80 mg ml$^{-1}$. The animals were kept in pletysmograph boxes at 37 °C and ventilated in parallel with 100% oxygen at a frequency of 40 breaths per minute and an inspiration/expiration ratio 1:1. Animals receiving the same dose of Curosurf served as positive and non-treated littermates as negative controls. Animals were ventilated with a standard pressure sequence of 35/0 (peak-insufflation pressure [cmH$_2$O]/PEEP [cmH$_2$O]) for 1 min, 23/3 for 15 min, 18/3 for 5 min, 13/3 for 5 min and 23/3 for 5 min. Finally, the lungs were ventilated for additional 5 min with nitrogen at 23/3 cm H$_2$O and then excised for LGV measurements using the water displacement technique.

**Lung histology.** The lungs were fixed by immersion in 4% neutral formalin, dehydrated and embedded in paraffin. Transverse sections were stained with hematoxylin and eosin. Alveolar volume density was measured with a computer-aided image analyzer using total parenchyma as reference volume.

**Statistics and animal models.** We used pregnant New Zealand White rabbits and the premature foetuses were delivered by caesarean on day 27 (term 31 days). Surfactant preparations were compared using 7–12 rabbit foetuses in each group. Exclusion criteria for the whole experiment were: mean weights of all foetuses, < 20 g or > 40 g; non-treated controls, mean tidal volume at 5 min > 5.5 ml kg$^{-1}$; positive controls (Curosurf-treated), mean tidal volume at 5 min < 11 ml kg$^{-1}$. Exclusion criteria for specific animals were: weight, < 20 g or > 40 g; pneu-mothorax. The experiments were not blinded. Instead, a rolling schedule was used, indicating that the order may differ from litter to litter. Newman–Keuls method for multiple comparisons was used for statistical analyses of all groups. The animal experiments were approved by the ethical committee (N198/12, Stockholms Norra Djurförsöksetiska Nämnd).

**Surface activity measurements.** Surface activity was measured using a captive bubble surfactometer (CBS). The test chamber was filled with 10% sucrose in saline. Approximately 2 µl of surfactant, 10 mg ml$^{-1}$, was injected into the sample chamber and allowed to float by buoyancy to the agarose ceiling. After that an air

bubble was placed under the ceiling and surface tension was measured at different time intervals from the time when the bubble was inserted and resting in contact with the surfactant. After 5 mins of adsorption the sample chamber was sealed and the quasi-static cycling was initiated. During quasi-static cycling the bubble was compressed stepwise until the minimal surface tension was reached and thereafter expanded to the initial size. This manoeuvre was repeated five times, and minimum and maximum surface tension ($\gamma_{min}$ and $\gamma_{max}$) as well as compression needed to reach a surface tension of $5\,mN\,m^{-1}$ were recorded and presented as median values from three experiments.

**Purification of rfhSP-D and rCCK-58.** Cells were thawed and centrifuged at 7,000$g$, 4 °C for 40 min. The supernatants were removed and the pellets were resuspended in 20 mM Tris, pH 8. The suspension buffer used for rfhSP-D was also supplemented with 1 mM CaCl$_2$. The fusion proteins were solubilized by sonication at 80% amplitude, 1 s on and 1 s off for a total of 2 min and were purified on Ni-Sepharose columns as described above. 20 mM Tris, 5 mM imidazole, pH 8 was used as washing buffer to remove contaminants. The fusion proteins were eluted with 20 mM Tris, 300 mM imidazole, pH 8 in 1 ml fractions. The absorbance at 280 nm was measured for each fraction, and protein-containing fractions were pooled. Imidazole was removed by over-night dialysis at 4 °C and in 5 l 20 mM Tris, pH 8, using Spectra/Por dialysis membranes with a 6–8 kDa molecular weight cut-off. Digestion was performed at 4 °C for 6 h using 2.5 mg 3C protease to 25 mg fusion protein at a concentration of 2 mg ml$^{-1}$ and in the presence of 1 mM DTT. After cleavage, a buffer exchange was performed to remove DTT prior to the second purification step. rfhSP-D was dialyzed over-night as described above and rCCK-58 was passed over a PD-10 desalting column (GE Healthcare). The samples were again loaded to Ni-Sepharose columns to bind the histidine tagged 3C protease and NT*. Flow-through was discarded since only a small amount of truncated rfhSP-D passed directly through the column and all of rCCK-58 remained unspecifically bound. Full-length rfhSP-D could be eluted using 20 mM Tris, 150 mM NaCl, 5 mM imidazole, pH 8 without contamination from the more strongly bound NT* and 3C protease. rCCK-58 bound much stronger and first, NT* and 3C protease were eluted using 20 mM Tris, 300 mM imidazole, pH 8. Second, pure rCCK-58 was eluted by stripping the column using 20 mM Tris, 100 mM EDTA, pH 8. Eluted fractions were pooled and the proteins were dialyzed over-night as described above. The proteins were concentrated to around 1.5 mg ml$^{-1}$ using Vivaspin 20 centrifugal tubes with a 5 kDa (rfhSP-D) or 3 kDa (rCCK-58) molecular weight cut-off (GE Healthcare).

**Size-exclusion chromatography of rfhSP-D.** Purified rfhSP-D was analysed by SEC using a Superdex 200 column equilibrated with 20 mM Tris, 150 mM NaCl, 1 mM EDTA, pH 8. 200 µl of protein sample (1.5 mg ml$^{-1}$) was loaded onto the column using a flow-rate of 0.5 ml min$^{-1}$. Eluted protein was detected by measuring optical absorbance at 280 nm. Molecular weight standards ferritin (440 kDa), aldolase (158 kDa), conalbumin (75 kDa), ovalbumin (43 kDa), carbonic anhydrase (29 kDa) and ribonuclease A (13.7 kDa) (GE Healthcare) were used for calibration and eluted at 8.56, 10.65, 12.06, 12.96, 14.26 and 15.64 ml, respectively. In a separate experiment, the trimeric protein was separated using the same conditions. Two 0.5 ml fractions were collected around an elution volume of 11.5 ml, where the peak had the highest intensity, and the sample was concentrated. Comparable amounts of proteins were taken from samples before and after SEC separation for SDS-PAGE analysis using a 15% acrylamide gel stained with Coomassie Brilliant blue.

**ESI-MS analysis of rfhSP-D.** SEC separated rfhSP-D protein was reconstituted into 100 mM ammonium acetate, pH 7.5 using biospin buffer exchange columns (Bio-Rad Laboratories). The sample was introduced into the mass spectrometer by gold-coated borosilicate needles produced in-house. Ion mobility analysis was performed on a Synapt 1T-wave mass spectrometer (Waters) operated in ToF mode. The settings were: capillary voltage, 0.8–1.3 kV; sample cone 80 V; source temperature, 20 °C; cone gas, off; trap collision energy, 5 V; transfer collision energy, 5 V; trap DC bias 8 V; backing pressure 6.8e0 mBar. Data was analysed using MassLynx 4.1 software (Waters).

**Data availability.** All relevant data are available from the authors. The rSP-C33Leu solution structure has been deposited at the Research Collaboratory for Structural Bioinformatics Protein Data Bank (PDB) with the accession code 5NDA.

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

## Acknowledgements

We thank Prof Howard Clark and Dr Alastair Watson at the University of Southampton, UK, for advice and for providing the rfhSP-D gene construct. This work was funded by the Swedish Research Council, FORMAS, Vinnova, VIAA Latvia NFI/R/2014/023 Grant and the InnovaBalt Project at Latvian Institute of Organic Synthesis.

## Author contributions

A.R. and J.J. conceived and designed the study; N.K., M.S., A.L., L.S. and K.N. performed experimental work related to cloning, protein expression, solubility analysis, purification and characterization; M.L. performed ESI-MS; T.C. performed *in vitro* surface activity measurements and *in vivo* experiments with rSP-C33Leu; M.O. performed HSQC-NMR analysis of NT; L.V. performed NMR structure determination; B.P. performed ion mobility measurements; P.P. performed negative staining TEM; H.B. cloned and purified

the NT*-rAβ1-40/rAβ1-42 constructs; Z.T. designed and cloned the MBP-rCCK-58 plasmid; J.J., A.R., N.K., N.P., H.J., K.J., H.H. and T.C. contributed by supervision and data analysis. N.K. wrote the main part of the manuscript with contributions from A.R. and J.J. All authors commented on the final version of the manuscript.

## Additional information

**Competing interests:** The authors declare no competing financial interests.

