## [Peer Review File · Nature Communications]

Reviewers' comments:

Reviewer #1 (Remarks to the Author):

The manuscript describes a fusion tag (NT*) for facilitating the expression and purification of peptides and proteins prone to aggregation. NT* was shown to be significantly more water-soluble than its corresponding wild type form from the N-terminal domain of a spider major ampullate silk protein. The expression levels of the two NT* fused peptides and one fused protein tested here were significantly higher than those of the pGB1 fused proteins (Fig. 4a-c). In this work, only two commercial tags (Trx and pGB1) were tested, and the performance of other widely used tags such as MBP, SUMO, and GST is unknown. The major ampullate silk protein contains three domains: N-terminal domain, C-terminal domain and repetitive domain. In fact, the repetitive domain is dominated by Gly (~37%), is relatively water-soluble (Huemmerich D, et al., *Biochemistry*, 2004, 43:13604), and is very different from transmembrane peptides and proteins. In addition, the three domains work together to ensure high solubility of a silk protein. If the NT* alone can sequester hydrophobic aggregates, any highly water-soluble tags shall work. So NT* may be just like one of the conventional tags. NT* could enhance the expression of the transmembrane peptides and the aggregation-prone protein tested here, but may not work for other peptides and proteins. This is similar to what happened to Trx and PGB1 – PGB1 enhanced the expression level of SP-C33Leu much more significantly than Trx. In addition, peptides smaller than 35 residues are synthesized chemically in a cost effective way nowadays, which are readily available from many commercial companies. If NT* can be shown to improve the expression and purification of transmembrane proteins, this tag will be more advantageous in general. Therefore, the novelty of this work is limited.

Protein purification by salt precipitation is a conventional method. Nevertheless, a combination of the precipitation and ethanol extraction methods was quite efficient for the purification of the two similar peptides studied here. It is difficult to predict if this method would become a general approach.

Detailed characterization of NT* did not provide new insights since previous studies from the same group (*Nat. commun.*, 5:3254) have already shown that the D40K mutant forms only monomers in the entire pH range and the K65D mutant exists mainly in monomers at pH>5.5. The structure of the peptide was solved in an organic solvent which is totally different from a bio-membrane and thus may not be functionally relevant. Functional study showed the recombinant peptide is active. This is expected since this peptide has the same sequence as the previously established one.

Below are some minor points:

1. Can rSP-C33Leu and rKL4 be purified using a Ni-sepharose column?
2. P.23, line 572, what was the final pH after adding HCl?
3. Figure 4, the peptides are smaller than 5 kDa. Why do they appear at a position of ~10 kDa?
4. Figure 4, the purified rfhSP-D still contained 2 bands with a ratio of ~3.

Reviewer #2 (Remarks to the Author):

The work of Kronqvist et al. deals with the recombinant expression of aggregation prone surfactant proteins using a highly soluble domain from a silk protein from spiders. The authors succeeded in the expression of the targets in *E. coli* in a soluble form. The NT* fusion tag could lead to interesting biotechnological applications. I have some concerns about the overall implications of the study and some methodological procedures:

- 1) Title and other parts of the manuscript: I think the authors generalize the usefulness of the NT* tool too much. They tested 2 TM peptides of the same family and another surfactant protein, and so "efficient protein production... (Title)" strikes me as too general and also diminishes the important result of obtaining rSP-C33Leu in a soluble active way. Maybe something like "Soluble human surfactant proteins production inspired by..." can be considered.
- 2) Abstract: First use of E. coli, change to Escherichia
- 3) Intro: "and are the targets of >50% of currently available pharmaceutical drugs". Please provide citation for this claim.
- 4) Intro: "recombinant NT is expressed at remarkably high levels in E. coli and purified protein can be concentrated to ~310 mg/mL before entering a gel state". Please provide citation for this claim.
- 5) Intro: The last sentence of the second paragraph (Considering that wild-type NT...) describes results and should be rephrased or removed. It disrupts the flow of the text.
- 6) "NMR spectra for NT* at both pH 7.2 and pH 5.5 are comparable to the spectrum measured for monomeric NTwt at pH 7.2 (Fig. 3a-c)". It is apparent that NT/NT* behave the same at pH 7.2 and 8.0. But why did the authors use a pH of 7.2 in the NMR experiments and pH 8.0 for the SEC experiments?
- 7) CD experiments: "in contrast to NTwt that has a lower refolding capacity, in particular at low pH (Supplementary Fig. 1b)." The differences in molar ellipticity before and after heating for NT do not look so different so as to make that claim, besides the final spectrum is a smoothed average. Is there a way to extract some value out of the spectra and make a quantitative assessment of refolding capability? The claim is too subjective (the authors state that NTwt has a lower refolding capacity, it does not look that way to me).
- 8) Tag comparison: "and exceeding those observed for PGB1 and Trx fusions (Supplementary Fig. 3a-c)". Please rephrase. It sounds like many Trx fusions were tested too, when in fact this was tested in one case only.
- 9) "In addition, they were highly stable and did not show signs of degradation as observed for Trx-rSP-C33Leu (Supplementary Fig. 3a)." Although there is an extra band below the fusion in Supp. Fig 3a, the authors did not actually test that this is in fact a degradation product. Please remove sentence or perform a Western blot with anti-His or anti-Trx antibodies.
- 10) The font changes to Cambria in the last sentence of the CCS results.
- 11) "Animals receiving the same dose of Curosurf® served as positive and non-treated littermates as negative controls." Is "same dose" referring to only the SP-C component or both SP-B + SP-C in Curosurf? Please clarify. Also, why the use of non-treated littermates as negative controls? It could be argued that the best negative control would have been fetuses treated with the phospholipid mixture alone.
- 12) "The purified 18.7 kDa protein migrated as expected on SDS-PAGE (Fig. 4f)" Two bands can be seen in the gel. Have the authors determined which one is the full length protein? It can also be a contaminant. In figure 8b, the arrow points at the lower band. In that figure, the molecular weights of the markers were not disclosed but, assuming it is the same as in figure 4f, why does the band is positioned right next to the 15 kDa band in 8b and above it in figure 4f?
- 13) "This confirmed that the main part of rfhSP-D adopts the 57 kDa trimer conformation that is essential for activity (Fig. 8c, Supplementary Fig. 10)." The spectrum is quite noisy. Only two peaks for deconvolution? Maybe more protein can be injected into the mass spec to get a reliable result. SAXS would be a more suitable technique in this case.
- 14) Discussion: "all peptides/protein were produced in a soluble and functional form after removal of the tag." Actually, the function of only one (SP-C33Leu) was tested. Rephrase. Also, there is an extra space after that sentence.
- 15) "...would allow them to arrange into micelle-like particles...". This represents the basis of the NT* tool but it was tested in only one case. TEM images showing the micelles for the other two fusions should be presented. That would make a strong case for the molecular mechanism of NT* effect on solubility.
- 16) "NT* allows for efficient purification of hydrophobic target peptides using just simple NaCl precipitation and ethanol extraction steps." Salt/ethanol precipitations are also proper conditions

for LPS purification. Have the authors considered that the final product may contain high levels of endotoxins? If I understood correctly, the animals were euthanized after 36 min for lung retrieval which may not be enough to detect adverse reactions. LPS contamination by the purification protocol of the authors should be discussed in the text. Better still, the amount of LPS could also be quantified.

17) Methods: "The previously described vector pT7HisTrxHisNT, containing the NTwt sequence"
Please add citation. Also specify here the species of the spider from where the NT gene was obtained.

18) Mutagenesis: Why did the authors digested, purified and recloned the NT* gene? The QuikChange kit allows for direct transformation of the mutated plasmid.

19) "(IPTG) to a final concentration of 0.5 mM and further incubation at 30 oC, 180 rpm for 4 hours." Maybe "...0.5 mM and the culture was further incubated at..."

20) Please state working concentrations of lysozyme, DNase and MgCl₂.

21) IMAC is not defined in the text at its first use.

22) Please state the concentration of thrombin in the dialysis bag. Also, it seems that thrombin is a contaminant in the final preparations of NTs. Could it interfere with some of the assays?

23) Why did the authors start using sonication when disrupting cells producing the fusion proteins when they had previously used the gentler lysozyme protocol for NT purification?

24) "Size-exclusion chromatography of NT*-SP-C33Leu" Add r before SP.

25) Extra space after "CYANA macro".

26) "...(POPG) 68:31 (w/w) at a concentration of 80 mg/mL." wt./wt. was used before. Check consistency.

27) "...mean value at 5 min > 5.5 mL/kg; positive controls (Curosurf-treated), mean value at 5 min < 11 mL/kg". Which value? I am guessing tidal volume?

28) "Approximately 2 μL of surfactant" Missing symbol.

29) ESI-MS analysis of rfhSP-D. Justify paragraph (is right aligned).

30) Figure 1a. "Repetitive region" next to the drawing.

Replies to reviewers' comments are formatted in bold.

Reviewer #1 (Remarks to the Author):

The manuscript describes a fusion tag (NT*) for facilitating the expression and purification of peptides and proteins prone to aggregation. NT* was shown to be significantly more water-soluble than its corresponding wild type form from the N-terminal domain of a spider major ampullate silk protein. The expression levels of the two NT* fused peptides and one fused protein tested here were significantly higher than those of the pGB1 fused proteins (Fig. 4a-c). In this work, only two commercial tags (Trx and pGB1) were tested, and the performance of other widely used tags such as MBP, SUMO, and GST is unknown.

This is a good point and in the revised version we have now included the expression and purification of several other peptides and proteins in fusion with NT*, including the cholecystokinin-58 peptide hormone (CCK-58), two amyloid β peptides (A β 1-40 and A β 1-42), a designed β -sheet protein (β 17), the human antimicrobial cathelicidin LL-37 precursor protein (hCAP18) and a native-like SP-C peptide (Supplementary Fig. 2). To further demonstrate the superior performance of NT* over conventional tags we evaluated also the MBP solubility tag in fusion with CCK-58 (Fig. 4c,g, Supplementary Fig. 3c and 4c). In addition, Supplementary Table 1 has been extensively revised to include the larger set of now analyzed proteins/peptides and to compare the yields of NT fusion proteins with published yields using alternative solubility tags. The introduction has also been revised in order to emphasize the larger number of proteins and peptides that have now been investigated.

The major ampullate silk protein contains three domains: N-terminal domain, C-terminal domain and repetitive domain. In fact, the repetitive domain is dominated by Gly (~37%), is relatively water-soluble (Huemmerich D, et al., Biochemistry, 2004, 43:13604), and is very different from transmembrane peptides and proteins. In addition, the three domains work together to ensure high solubility of a silk protein. If the NT* alone can sequester hydrophobic aggregates, any highly water-soluble tags shall work.

The reviewer raises an interesting point. We agree that the MaSp1 repetitive domain is not comparable to transmembrane (TM) peptides in terms of hydrophobicity. Nevertheless, it has been proposed that silk proteins form micelles with the repetitive domain sequestered by the terminal domains (see refs 39-41 in the manuscript). The expression of TM peptides in fusion with NT was a hypothesis based on these observations. In theory, any highly soluble tag may have the ability to sequester hydrophobic aggregates but to our knowledge, this has not previously been described in the literature.

So NT* may be just like one of the conventional tags. NT* could enhance the expression of the transmembrane peptides and the aggregation-prone protein tested here, but may not work for other peptides and proteins. This is similar to what happened to Trx and PGB1 – PGB1 enhanced the expression level of SP-C33Leu much more significantly than Trx.

As described above, we have studied several more proteins and peptides in the

revised version, and according to both our own experimental results and published data, the total yields of NT* fusion proteins are significantly higher compared to conventional tags in all cases. This is now described in Results (p.10-11) and Discussion (p. 17). Still, we want to emphasize that no tag can be expected to improve the expression and solubility of all types of proteins. This must be evaluated empirically and therefore it is highly beneficial that new tags are developed and prove to be efficient for peptides and proteins that previously have been refractory to recombinant production or have not been produced at sufficient yields.

In addition, peptides smaller than 35 residues are synthesized chemically in a cost effective way nowadays, which are readily available from many commercial companies. **The reviewer has a point regarding the decreasing costs for chemical synthesis of shorter peptides, but even today, recombinant techniques are substantially less expensive when large production yields are required. Moreover, as mentioned in the introduction part, a well-established protocol for recombinant expression of a peptide also allows for efficient screening of new analogues with improved functionality, which is another important aspect for research and development.**

If NT* can be shown to improve the expression and purification of transmembrane proteins, this tag will be more advantageous in general. Therefore, the novelty of this work is limited.

The restraining aspect for soluble expression and purification of transmembrane proteins in general is the presence of a hydrophobic transmembrane domain. In the revised version, we now show that also native SP-C, which is one of the most hydrophobic TM peptides known, can be purified in fusion with NT*. This is highly encouraging for future investigation of larger transmembrane proteins.

Protein purification by salt precipitation is a conventional method. Nevertheless, a combination of the precipitation and ethanol extraction methods was quite efficient for the purification of the two similar peptides studied here. It is difficult to predict if this method would become a general approach.

The high hydrophobicity of TM peptides will generally allow them to precipitate in a high-salt solution and subsequently dissolve in a non-polar solvent, and it is therefore fair to conclude that this could become a general method for this family of peptides. However, the method would likely not be applicable for all peptides and in particular not for those which are water-soluble. To make this clearer, the word “peptide” has been changed to “TM peptide” in the headings describing this method in Results (p. 12) and Methods (p. 25). Although the method is conventional, the novelty of the protocol lies in the combination with the NT solubility tag, since it is unusual for all CNBr cleaved fragments of a protein to all be insoluble in a non-polar solvent and this is what allows for separation of pure TM-peptide from the otherwise contaminating fragments of the solubility tag. This is mentioned in Results (p. 12).

Detailed characterization of NT* did not provide new insights since previous studies from the same group (Nat. commun., 5:3254) have already shown that the D40K mutant

forms only monomers in the entire pH range and the K65D mutant exists mainly in monomers at pH>5.5.

The characteristics of double or multiple mutants cannot be expected to represent those of single mutants in an additive manner. In this case particularly, the net charge-neutral swap of D40K and K65D could potentially have introduced a novel salt-bridge that would have promoted dimer formation, in analogy to the mechanism observed for the wt protein. This is in contrast to the previously studied single mutants referred to by the reviewer, which both changed the net charge and introduced repulsive K-K or D-D intersubunit interactions. The Trp fluorescence data in this manuscript indeed show that the double mutant remains monomeric and we support this finding with HSQC NMR and size exclusion chromatography, methods that have not previously been used to characterize any D40 or K65 mutants. For this reason we respectfully disagree with the reviewer and we find the characterization of NT* to be important for the understanding of the behavior of this mutant.

The structure of the peptide was solved in an organic solvent which is totally different from a bio-membrane and thus may not be functionally relevant. Functional study showed the recombinant peptide is active. This is expected since this peptide has the same sequence as the previously established one.

The NMR structure for native SP-C in the same organic solvent mixture as now used for rSP-C33Leu has been shown to well represent the structure in lipid environments, as analyzed by NMR, FTIR and CD spectroscopy (Johansson et al., Biochemistry, 1994; Johansson et al., FEBS Lett., 1995; Vandenbussche et al., Eur. J. Biochem., 1992). This is now pointed out in the manuscript on p. 14. We also compare the rSP-C33Leu structure with that of native SP-C and find them to be very similar, which supports that rSP-C33Leu is a useful analogue for formulation if synthetic surfactants and that the new expression and purification methods in our manuscript do not cause any inadvertent conformational changes.

Below are some minor points:

1. Can rSP-C33Leu and rKL4 be purified using a Ni-sepharose column?

The purification of rSP-C33Leu and rKL4 using a Ni-sepharose column is described in the manuscript under the heading “Comparison of different solubility tags” in Results (p. 10-11). Text has now been added under the heading “Purification of fusion proteins for comparison of yields” in Methods (p. 24) to clarify that all fusion proteins have been evaluated by IMAC purification in order to directly compare expression and solubility of different tags in a similar manner. Later on, rSP-C33Leu and rKL4 peptides were purified using the more efficient salt precipitation/ethanol extraction protocol prior to further characterization.

2. P.23, line 572, what was the final pH after adding HCl?

The final pH used during CNBr cleavage was 1.0. This has been added in Methods (p. 26).

3. Figure 4, the peptides are smaller than 5 kDa. Why do they appear at a position of

~10 kDa?

It is rather common that the size of a peptide or protein does not fully correlate with the apparent size determined by SDS-PAGE. ESI-MS confirmed that rSP-C33Leu has the correct size although it migrates differently in the gel.

4. Figure 4, the purified rfhSP-D still contained 2 bands with a ratio of ~3.

The presence of two bands is probably an SDS-PAGE artifact reflecting some preserved SDS-stable conformational difference between two populations of the protein. This conclusion is supported by the observation that both bands appear with the same intensity after narrow separation of the SEC peak corresponding to the trimer, (Fig. 8a-b) showing that they are both functional and migrate with the same hydrodynamic radius on SEC. A clarification has been added to the manuscript in Results (p. 16-17).

Reviewer #2 (Remarks to the Author):

The work of Kronqvist et al. deals with the recombinant expression of aggregation prone surfactant proteins using a highly soluble domain from a silk protein from spiders. The authors succeeded in the expression of the targets in *E. coli* in a soluble form. The NT* fusion tag could lead to interesting biotechnological applications. I have some concerns about the overall implications of the study and some methodological procedures:

1) Title and other parts of the manuscript: I think the authors generalize the usefulness of the NT* tool too much. They tested 2 TM peptides of the same family and another surfactant protein, and so “efficient protein production... (Title)” strikes me as too general and also diminishes the important result of obtaining rSP-C33Leu in a soluble active way. Maybe something like “Soluble human surfactant proteins production inspired by...” can be considered.

This important point was also raised by reviewer #1, and we have now included the expression and purification of several other peptides and proteins in fusion with NT*, including the cholecystokinin-58 peptide hormone (CCK-58), two amyloid β peptides (A β 1-40 and A β 1-42), a designed β -sheet protein (β 17), the human antimicrobial cathelicidin LL-37 precursor protein (hCAP18) and a native-like SP-C peptide (Supplementary Fig. 2) To further demonstrate the improved performance of NT* over conventional tags we evaluated also the MBP solubility tag in fusion with CCK-58 (Fig. 4c,g, Supplementary Fig. 3c and 4c). In addition, Supplementary Table 1 has been extensively revised to include the larger set of analyzed proteins/peptides and to compare the yields of NT fusion proteins with published yields using alternative solubility tags. The results support that NT* is useful for a larger set of proteins and peptides and we therefore prefer to keep the title unchanged.

2) Abstract: First use of *E. coli*, change to *Escherichia*

The first use of *Escherichia* has been moved from Introduction to Abstract (p. 2).

3) Intro: “and are the targets of >50% of currently available pharmaceutical drugs”.

Please provide citation for this claim.

A reference has been added and the percentage has been changed to ~60% to more precisely cite the publication (p. 3).

4) Intro: “recombinant NT is expressed at remarkably high levels in *E. coli* and purified protein can be concentrated to ~310 mg/mL before entering a gel state”. Please provide citation for this claim.

This claim refers to data in Hedhammar et al. 2008 and to unpublished data, which are later compared to the concentration of the NT* mutant in Results (p. 9). A reference to Hedhammar et al. 2008 and a “data not shown” statement has been added (p. 5) to clarify that this statement partly refers to our current results. Although results should be avoided in the Introduction part, we think this statement is important for the understanding of the rationale behind the attempt to use NT as a solubility tag.

5) Intro: The last sentence of the second paragraph (Considering that wild-type NT...) describes results and should be rephrased or removed. It disrupts the flow of the text.

We fully agree with the reviewer that this information is not appropriate as part of the introduction. The sentence has been rephrased and moved to Results (p. 8).

6) “NMR spectra for NT* at both pH 7.2 and pH 5.5 are comparable to the spectrum measured for monomeric NT_{wt} at pH 7.2 (Fig. 3a-c)”. It is apparent that NT/NT* behave the same at pH 7.2 and 8.0. But why did the authors use a pH of 7.2 in the NMR experiments and pH 8.0 for the SEC experiments?

The use of pH 8 is the standard choice for most experiments since NT_{wt} then reach a fully monomeric state according to Trp fluorescence spectra in the absence of salt (Fig. 2a-b and Kronqvist et al., Nat. Commun., 2014). However, obtaining HSQC NMR spectra at high pH gives signal attenuation due to rapid exchange between amide hydrogen atoms and water. For this reason, NMR experiments were performed pH 7.2 in the presence of 150 mM NaCl, which gives the same NT monomer-dimer as at pH 8 (Kronqvist et al., Nat. Commun., 2014). A sentence describing this has been added to the manuscript (p. 9).

7) CD experiments: “in contrast to NT_{wt} that has a lower refolding capacity, in particular at low pH (Supplementary Fig. 1b).” The differences in molar ellipticity before and after heating for NT do not look so different so as to make that claim, besides the final spectrum is a smoothed average. Is there a way to extract some value out of the spectra and make a quantitative assessment of refolding capability? The claim is too subjective (the authors state that NT_{wt} has a lower refolding capacity, it does not look that way to me).

As the reviewer observed, there is no qualitative change of the CD spectra after refolding and hence quantitative assessment of secondary structure contents by e.g. deconvolution would not result in differences before and after refolding. However, the signal intensity (i.e. amplitude), which is dependent on the protein concentration, is reproducibly about 15% lower between 205-225 nm after refolding of NT_{wt} at low pH, which reveals that some of the protein has precipitated due to a lower capacity to refold. A comment has been added to

clarify how this conclusion was drawn (p. 10).

8) Tag comparison: “and exceeding those observed for PGB1 and Trx fusions (Supplementary Fig. 3a-c)”. Please rephrase. It sounds like many Trx fusions were tested too, when in fact this was tested in one case only.

We agree that this sentence could be misinterpreted. In the revised version we also have included comparisons with MBP and other tags and the sentence has been changed to “and exceeding those observed for other fusion tags used for direct comparison” (p. 10-11).

9) “In addition, they were highly stable and did not show signs of degradation as observed for Trx-rSP-C33Leu (Supplementary Fig. 3a).” Although there is an extra band below the fusion in Supp. Fig 3a, the authors did not actually test that this is in fact a degradation product. Please remove sentence or perform a Western blot with anti-His or anti-Trx antibodies.

The experimental work made it quite clear to us that the lower band increased with time, which is expected from a degradation product. However, we fully agree that this is not obvious to the reader based on the presented data and the sentence has therefore been removed (p. 11).

10) The font changes to Cambria in the last sentence of the CCS results.

The font has been changed to Times New Roman (p. 14).

11) “Animals receiving the same dose of Curosurf® served as positive and non-treated littermates as negative controls.” Is “same dose” referring to only the SP-C component or both SP-B + SP-C in Curosurf? Please clarify. Also, why the use of non-treated littermates as negative controls? It could be argued that the best negative control would have been fetuses treated with the phospholipid mixture alone.

The same dose refers to the total amounts of phospholipids administered and this has been clarified in Results (p. 15). As the reviewer suggests the phospholipid mixture alone is indeed a good negative control and we have previously shown that treatment with phospholipids only gives no improvement in tidal volumes or lung gas volumes compared to non-treated controls (Calkovska et al., Neonatology, 2016). A sentence describing this (including the citation) has been added in Results (p. 15).

12) “The purified 18.7 kDa protein migrated as expected on SDS-PAGE (Fig. 4f)” Two bands can be seen in the gel. Have the authors determined which one is the full length protein? It can also be a contaminant. In figure 8b, the arrow points at the lower band. In that figure, the molecular weights of the markers were not disclosed but, assuming it is the same as in figure 4f, why does the band is positioned right next to the 15 kDa band in 8b and above it in figure 4f?

We are grateful for this observation, which was also made by reviewer #1. The presence of two bands is probably an SDS-PAGE artifact reflecting some preserved SDS-stable conformational difference between two populations of the protein. This conclusion is supported by the observation that both bands appear with the same intensity after narrow separation of the SEC peak corresponding to

the trimer, (Fig. 8a-b) showing that they are both functional and migrate with the same hydrodynamic radius on SEC, and clarifications have been added to the manuscript in Results (p. 16-17). Regarding the positioning of the bands relative to the protein marker, the migration profile did differ slightly between experiments depending on the applied voltage and how far the proteins were allowed to migrate.

13) “This confirmed that the main part of rfhSP-D adopts the 57 kDa trimer conformation that is essential for activity (Fig. 8c, Supplementary Fig. 10).” The spectrum is quite noisy. Only two peaks for deconvolution? Maybe more protein can be injected into the mass spec to get a reliable result. SAXS would be a more suitable technique in this case.

Narrow charge state distributions are highly indicative of folded proteins, and considering the gentle ionization conditions employed, it is reasonable to conclude that the dominant charge state series in the spectrum in Supplementary Fig. 10 represents the intact SP-D trimer. Furthermore, as highlighted by the grey box, the neighboring 17+ (m/z 3330) charge state (unique for the trimer) as well as the 14+ (m/z 4045) charge state (broad peak indicative of salt adducts) can be detected in the spectrum. These peaks were now labeled for clarity (Supplementary Fig. 10).

14) Discussion: “all peptides/protein were produced in a soluble and functional form after removal of the tag.” Actually, the function of only one (SP-C33Leu) was tested. Rephrase. Also, there is an extra space after that sentence.

Since we fully agree with this comment, now that we also included several more proteins and peptides that are less characterized, the statement about functionality has been removed (p. 17).

15) “...would allow them to arrange into micelle-like particles...”. This represents the basis of the NT* tool but it was tested in only one. TEM images showing the micelles for the other two fusions should be presented. That would make a strong case for the molecular mechanism of NT* effect on solubility.

In the revised version, we have added a TEM image showing that also NT*-rKL4 arrange into micelle-like particles of a similar radius compared to NT*-rSP-C33Leu (Results, p. 12, Supplementary Fig. 5). However, micelle formation is not expected to be the underlying mechanism for solubilizing hydrophilic peptides and proteins, but rather, this is likely mediated by the high solubility and folding capacity, as observed for conventional tags. This is now emphasized in the Discussion (p. 18).

16) “NT* allows for efficient purification of hydrophobic target peptides using just simple NaCl precipitation and ethanol extraction steps.” Salt/ethanol precipitations are also proper conditions for LPS purification. Have the authors considered that the final product may contain high levels of endotoxins? If I understood correctly, the animals were euthanized after 36 min for lung retrieval which may not be enough to detect adverse reactions. LPS contamination by the purification protocol of the authors should be

discussed in the text. Better still, the amount of LPS could also be quantified.

We are aware of this potential problem and now point it out in the manuscript. We have performed similar animal experiments as described in the manuscript for four hours and still see no negative effects (p. 15-16). This supports that the rSP-C33Leu preparations as described contain low LPS amounts, We are planning to describe these data and further results on LPS levels in future articles.

17) Methods: "The previously described vector pT7HisTrxHisNT, containing the NTwt sequence" Please add citation. Also specify here the species of the spider from where the NT gene was obtained.

Citation and species have been added in Methods (p. 19).

18) Mutagenesis: Why did the authors digested, purified and recloned the NT* gene? The QuikChange kit allows for direct transformation of the mutated plasmid.

The full vector never becomes sequenced and although the QuickChange kit allows direct transformation, we routinely sub-clone merely as a precaution to avoid random mutations over time.

19) "(IPTG) to a final concentration of 0.5 mM and further incubation at 30 oC, 180 rpm for 4 hours." Maybe "...0.5 mM and the culture was further incubated at..."

20) Please state working concentrations of lysozyme, DNase and MgCl₂.

21) IMAC is not defined in the text at its first use.

Changes have been made according to the suggestions in points 19-21.

22) Please state the concentration of thrombin in the dialysis bag. Also, it seems that thrombin is a contaminant in the final preparations of NTs. Could it interfere with some of the assays?

Thrombin was added to the fusion protein at a 1/1000 ratio (w/w) and this is now also stated in Methods (p. 21). A contaminant at such low concentration is not detectable in any of the assays used.

23) Why did the authors start using sonication when disrupting cells producing the fusion proteins when they had previously used the gentler lysozyme protocol for NT purification?

Hydrophobic or aggregation prone target proteins interact with cell membranes and *E. coli* proteins, which naturally leads to a reduced solubility of the NT* fusion protein compared to NT* alone. We therefore introduced sonication, not only to disrupt the cells, but also in order to more efficiently break weaker protein complexes that otherwise would pull the fusion protein into the pellet fraction.

24) "Size-exclusion chromatography of NT*-SP-C33Leu" Add r before SP.

25) Extra space after "CYANA macro".

26) "...(POPG) 68:31 (w/w) at a concentration of 80 mg/mL." wt./wt. was used before. Check consistency.

27) "...mean value at 5 min > 5.5 mL/kg; positive controls (Curosurf-treated), mean value at 5 min < 11 mL/kg". Which value? I am guessing tidal volume?

- 28) "Approximately 2 μ L of surfactant" Missing symbol.
- 29) ESI-MS analysis of rfhSP-D. Justify paragraph (is right aligned).
- 30) Figure 1a. "Repetitive region" next to the drawing.

Changes have been made according to the suggestions in points 24-30.

REVIEWERS' COMMENTS:

Reviewer #1 (Remarks to the Author):

In the revised manuscript, to prove NT* is a better tag the authors added following data: MBP tag, one transmembrane peptide SP-Css that belongs to the same family of SP-C33Leu, and 4 other proteins or peptides (A β 1-40, A β 1-42, hCAP18, and β 17). NT* seems to be a better tag than MABP, pGB1 and Trx, but it may not be better than those untested tags. So it is over-claimed by stating that "NT*-transmembrane protein fusions yield up to eight times more of soluble protein in *Escherichia coli* than fusions with conventional tags" in the abstract and summary. The authors tested only one family of transmembrane peptides, but did not provide additional data for membrane proteins or transmembrane peptides.

Minor points:

Line 89, repeated alanine and glycine rich segments exist in major and minor ampullate silk.

Line 222, what was the condition for concentrating the protein to 570 mg/ml?

Reviewer #2 (Remarks to the Author):

The authors have satisfactorily addressed the reviewers queries. Of note, the authors have expanded the number of fusion proteins tested. By doing that, they strengthened the message of the manuscript.

I have no further questions.

Replies to reviewers' comments are formatted in bold.

Reviewer #1 (Remarks to the Author):

In the revised manuscript, to prove NT* is a better tag the authors added following data: MBP tag, one transmembrane peptide SP-Css that belongs to the same family of SP-C33Leu, and 4 other proteins or peptides (A β 1-40, A β 1-42, hCAP18, and β 17). NT* seems to be a better tag than MABP, pGB1 and Trx, but it may not be better than those untested tags. So it is over-claimed by stating that "NT*-transmembrane protein fusions yield up to eight times more of soluble protein in Escherichia coli than fusions with conventional tags" in the abstract and summary. The authors tested only one family of transmembrane peptides, but did not provide additional data for membrane proteins or transmembrane peptides.

In the revised version of the manuscript we directly compare NT* to MBP, PGB1 and Trx but also show the superior performance of NT* compared to tags that previously have been described in in the literature (including GST, SN, IFABP, (NANP)₁₉ and Ubiquitin). This has strengthened our claim, but we agree with the author that a few statements could be perceived as too general. We have therefore changed the statement in the abstract to "...up to eight times more of soluble protein in Escherichia coli than fusions with *several* conventional tags" and also a statement in the discussion on p. 19 to "We benchmark the performance of NT* to *several* commonly used tags..."

Minor points:

Line 89, repeated alanine and glycine rich segments exist in major and minor ampullate silk.

To clarify that this does not apply to all spider silk types, the sentence has been changed to "Silk from ampullate glands are built up from extensive stretches of repeated alanine- and glycine-rich..." on p. 4.

Line 222, what was the condition for concentrating the protein to 570 mg/ml?

The protein was concentrated by ultrafiltration in Vivaspin concentrator tubes and a description of the method has now been added under the new heading "Centrifugal filter concentration" in Methods on p. 21.

Reviewer #2 (Remarks to the Author):

The authors have satisfactorily addressed the reviewers queries. Of note, the authors have expanded the number of fusion proteins tested. By doing that, they strengthened the message of the manuscript.

I have no further questions.